# Climate warming and elevated $CO_2$ alter peatland soil carbon sources and stability

Nicholas O. E. Ofiti [1,2] ✉, Michael W. I. Schmidt [1], Samuel Abiven [2,3], Paul J. Hanson [4], Colleen M. Iversen [4], Rachel M. Wilson[5], Joel E. Kostka [6], Guido L. B. Wiesenberg [1] & Avni Malhotra[1,7]

Peatlands are an important carbon (C) reservoir storing one-third of global soil organic carbon (SOC), but little is known about the fate of these C stocks under climate change. Here, we examine the impact of warming and elevated atmospheric $CO_2$ concentration ($eCO_2$) on the molecular composition of SOC to infer SOC sources (microbe-, plant- and fire-derived) and stability in a boreal peatland. We show that while warming alone decreased plant- and microbe-derived SOC due to enhanced decomposition, warming combined with $eCO_2$ increased plant-derived SOC compounds. We further observed increasing root-derived inputs (suberin) and declining leaf/needle-derived inputs (cutin) into SOC under warming and $eCO_2$. The decline in SOC compounds with warming and gains from new root-derived C under $eCO_2$, suggest that warming and $eCO_2$ may shift peatland C budget towards pools with faster turnover. Together, our results indicate that climate change may increase inputs and enhance decomposition of SOC potentially destabilising C storage in peatlands.

The capacity of peatlands to store more than one-third of the global SOC pool under waterlogged conditions[1,2] has been touted as a powerful natural form of C sequestration[3]. However, there is concern that rising global temperatures and atmospheric $CO_2$ concentrations, along with corresponding changes in hydrology and biology, have the potential to destabilize SOC stocks and increase the flux of $CO_2$ and $CH_4$ from peat soils to the atmosphere[2,4], amplifying the drivers of climate change. Yet processes regulating the formation and stability of peat SOC, and their responses to interactive climate change factors remain unclear[5,6], making it difficult to forecast changes in C dynamics in peatland ecosystems[7]. In peatlands, SOC stability is controlled by a suite of environmental conditions, including temperature, hydrology (water saturation), and pH[8,9], and biotic factors such as plant and microbial community composition and functioning[10,11] as well as the chemical characteristics of the peat itself[12,13]. These biotic and abiotic factors regulate the current rates and pathways of enzymatically catalysed SOC

decomposition[8,10], and they are expected to determine the response of SOC decomposition under future environmental change[5,7].

A key question for future peatland C balance is whether a portion of SOC that is currently stable may become more accessible to microbial decomposition[2,4,7], via direct effects of rising temperature and $CO_2$ concentration, indirect effects of altered water table[9,14,15], or due to changes in plant production and thus litter inputs to SOC[11,16,17]. Soil warming, occurring more rapidly at high compared to low latitudes[18,19], can lead to lower water-tables and increases in peat aeration[9,15]. Changes in moisture availability combined with the fertilization effects of elevated $CO_2$ can alter plant biomass production and the quality of litter inputs[10,16,20]. Such a change in peat aeration and plant litter (including root exudates) inputs can impact the microbially-mediated SOC turnover because soil water content and litter chemistry regulate microbial community composition, function, and enzymatic activity[8–11]. Differences in litter chemistry can also affect

[1]Department of Geography, University of Zurich, Zurich, Switzerland. [2]CEREEP-Ecotron Ile De France, ENS, CNRS, PSL Research University, Saint-Pierre-lès-Nemours, France. [3]Laboratoire de Géologie, Département de Géosciences, Ecole normale supérieure (ENS), Paris, France. [4]Environmental Sciences Division and Climate Change Science Institute, Oak Ridge National Laboratory, Oak Ridge, TN, USA. [5]Department of Earth, Ocean and Atmospheric Sciences, Florida State University, Tallahassee, FL, USA. [6]School of Biological Sciences and School of Earth and Atmospheric Sciences, Center for Microbial Dynamics and Infection, Georgia Institute of Technology, Atlanta, GA, USA. [7]Present address: Biological Sciences Division, Pacific Northwest National Laboratory, Richland, WA, USA. ✉e-mail: nicholas.ofiti@geo.uzh.ch

the sensitivity of organic matter decomposition to soil temperature, with further effects on C cycling[10,21]. Consequently, if altered organic matter sources persist over extended periods, they are likely to stimulate the mineralisation of currently stable C[22]. In short, climate-driven changes in water-table and plant litter inputs are likely to modify C storage[5,6], but the underlying mechanisms and their interactions, particularly those resulting in shifts in organic matter inputs and composition are still unknown in peatlands[7]. The role of such plant-soil interactions in regulating future SOC storage merits attention because a much larger proportion of SOC may be susceptible to climate-mediated losses than previously assumed[7,19].

Here, we investigate the effects of warming and elevated atmospheric $CO_2$ concentration (e$CO_2$) on peatland SOC molecular composition to infer SOC sources, stability, and implications for storage. To describe SOC dynamics, we partitioned bulk soil C into classes of molecular compounds comprising SOC defined operationally by organic matter origins (plant-, microorganism- and fire-derived) and potential decomposition rates[23–25]. Specifically, we targeted SOC compounds with distinct turnover times. Firstly, we targeted solvent-extractable compounds (alkanoic acids, alkanols, alkanes, steroids, and terpenoids, derived from plant- and microbial-material[26]) that are thought to turn over predominantly on a timescale of bulk SOC or faster[24,27]. Secondly, we targeted SOC components from different plant origins, hydrolysable biopolymers distinct to either leaf/needle (cutin) or root/bark (suberin) compartments (and presumably slowly cycling)[26–29]. Lastly, we targeted lignin and pyrogenic carbon (PyC) representing the most slowly cycling C[23,30–32]. We assessed the stability of these SOC compound classes following 4 years of warming and 2 years of e$CO_2$ at an in-situ climate manipulation experiment (SPRUCE: Spruce and Peatland Responses Under Changing Environments). The SPRUCE experiment provides a powerful climate change gradient whereby above- and below-ground warming (whole-ecosystem warming; +0, +2.25, +4.5, +6.75, +9 °C above ambient) are crossed with ambient and elevated $CO_2$ concentrations (+500 ppm above ambient)[33]. Given that high-latitude soils are projected to experience temperature increases of up to $8.3 \pm 1.9$ °C by 2100 under a high $CO_2$ emission scenario (RCP8.5)[19], this unique experimental set up allowed us to explore the fate of SOC across mild to extreme future warming scenarios. From the perspective of long-term peatland C sequestration, we are particularly interested in the response of polymeric SOC components because kinetic theory predicts that decomposition and turnover of complex SOC molecules (greater chemical stability) should be more sensitive to temperature changes than simple molecules[34]. In peatlands, warming is typically accompanied by a lower water-table coupled with peat aeration, which can further impact the apparent temperature sensitivity of organic matter decomposition[8,9,14,15]. Consequently, if temperature and oxygenation effects predominate over other biotic and abiotic factors that influence decomposition dynamics[9,25,34], we hypothesize that complex SOC molecules such as lignin, and PyC that have been proposed to turn over much slower than bulk C in soils due to their polyaromatic chemical structure[23,30,35], would degrade faster. Overall, our study rejects this hypothesis that complex SOC molecules are more responsive to temperature changes. Instead, we found that all molecular compounds comprising SOC, regardless of source and complexity, were vulnerable to shifts in climate drivers, demonstrating the high sensitivity of peatlands to climate change.

## Results and discussion
### Warming and e$CO_2$ have divergent effects on plant- and microorganism-derived SOC
First, we assessed how whole-ecosystem warming under ambient $CO_2$ modified SOC molecular composition. Four years of experimental warming decreased SOC molecules content in the surface peat (0–30 cm; Fig. 1; Fig. S2). Solvent-extractable compounds of plant and

microbial origins[26,27,29] declined by 30% between the 0 and 9 °C treatments which corresponds to a slope of $-0.79 \pm 0.2$ mg g$^{-1}$ per °C increase in temperature ($r^2 = 0.60$, $p = 0.0004$; Fig. 1a, Fig. S2a). Similarly, plant-derived hydrolysable biopolymers that presumably cycle more slowly[26,28,36], decreased by 14% between the 0 and 9 °C treatments, corresponding to $-0.65 \pm 0.2$ mg g$^{-1}$ per °C increase in temperature ($r^2 = 0.52$, $p < 0.0001$; Fig. 1b, Fig. S2b), likely due to accelerated microbial decomposition of soil C[4,37]. Surprisingly, the concentration of lignin phenols, which are made up of phenolic units that are expected to decompose faster under aerobic conditions[8,9,15], instead increased significantly in the surface peat by 12% between the 0 and 9 °C treatments ($0.51 \pm 0.2$ mg g$^{-1}$ per °C warming; $r^2 = 0.18$, $p = 0.01$; Fig. 1d, Fig. S2d), indicating that warming might have induced lignin stabilization. Indeed, it is plausible that more lignin phenols entered soils via increased root litter production or that a shift in plant species composition toward shrubs richer in these compounds[16,17] may explain this increase. It is also possible that this apparent increase in lignin phenols may be due to accelerated decomposition of more rapidly cycled labile C given that warming has increased the availability of labile sugars (and protein) in this experiment[38]. In contrast, ratios of commonly used lignin oxidation (degradation) indices; acid-to-aldehyde ratios of vanillyl and syringyl phenols, which typically serve as proxies for oxidation[32], increased linearly with increasing temperatures (Fig. 2a-c), suggesting that warming, as expected, induced lignin degradation. The observed relative increase in lignin phenols (Fig.1d; Fig. S1) thus reflects a combination of increased organic matter inputs via plant litter and root exudates, enhanced degradation of more labile SOC molecules induced by higher temperatures[16,38] and a better resilience of these molecules to warming as compared to the more labile ones.

Overall, the above findings indicate that warming stimulated a loss of SOC molecules, irrespective of their origins and potential decomposition rates – although lignin phenols showed a lower response than expected. These transformations likely reflect extensive decomposition driven by low water tables during summer dry periods[17], compounded by warm temperatures and increased above- and below-ground plant productivity[16,17,20], and labile C inputs to SOC[38,39]. Warming is typically accompanied by a lower water-table and enhanced peat aeration, which can affect the temperature sensitivity of organic matter decomposition[8,9,14,15]. Here, warming resulted in a substantial water level drawn down during summer dry periods (30 cm below hollow surface in warmer plots)[17]. Since soil microbial activity is intrinsically temperature sensitive[5,40], the lowered water table, together with additional plant litter and labile C inputs[10,16,17,38] potentially stimulated microbial SOC transformation (priming), leading to measurable loss in SOC components.

We then assessed how warming combined with an elevated atmospheric $CO_2$ concentration (e$CO_2$) modified key SOC molecules. We tested the hypothesis that e$CO_2$ would result in increased plant litter production and C transfer from plant litter into SOC, thereby buffering warming-induced SOC losses as temperatures rise[17,41,42]. Our results support this hypothesis and provide evidence that the interaction of warmer temperatures and e$CO_2$ led to increased plant C inputs to SOC in the surface peat, potentially offsetting the warming-induced C losses (Fig. 1; Fig. S2b). We observed that concentrations of hydrolysable biopolymers under e$CO_2$ increased by 18% between the 0 and 9 °C treatments ($0.86 \pm 0.3$ mg g$^{-1}$ per °C warming; $r^2 = 0.42$, $p < 0.0001$; Fig. 1b; Fig. S2b) (indicating plant contribution), despite a non-significant increase in the concentration of solvent-extractable compounds (from $19.8 \pm 0.9$ mg g$^{-1}$ in the +0 °C plot to $23.3 \pm 1.1$ mg g$^{-1}$ in the +9 °C plot; $p > 0.05$; Fig. 1a). The source of hydrolysable biopolymers could be roots, as evidenced by elevated concentrations of root-specific suberin monomers (Fig. 3b) and root litter production[16]. Surprisingly, the concentration of lignin phenols decreased linearly with increasing temperatures

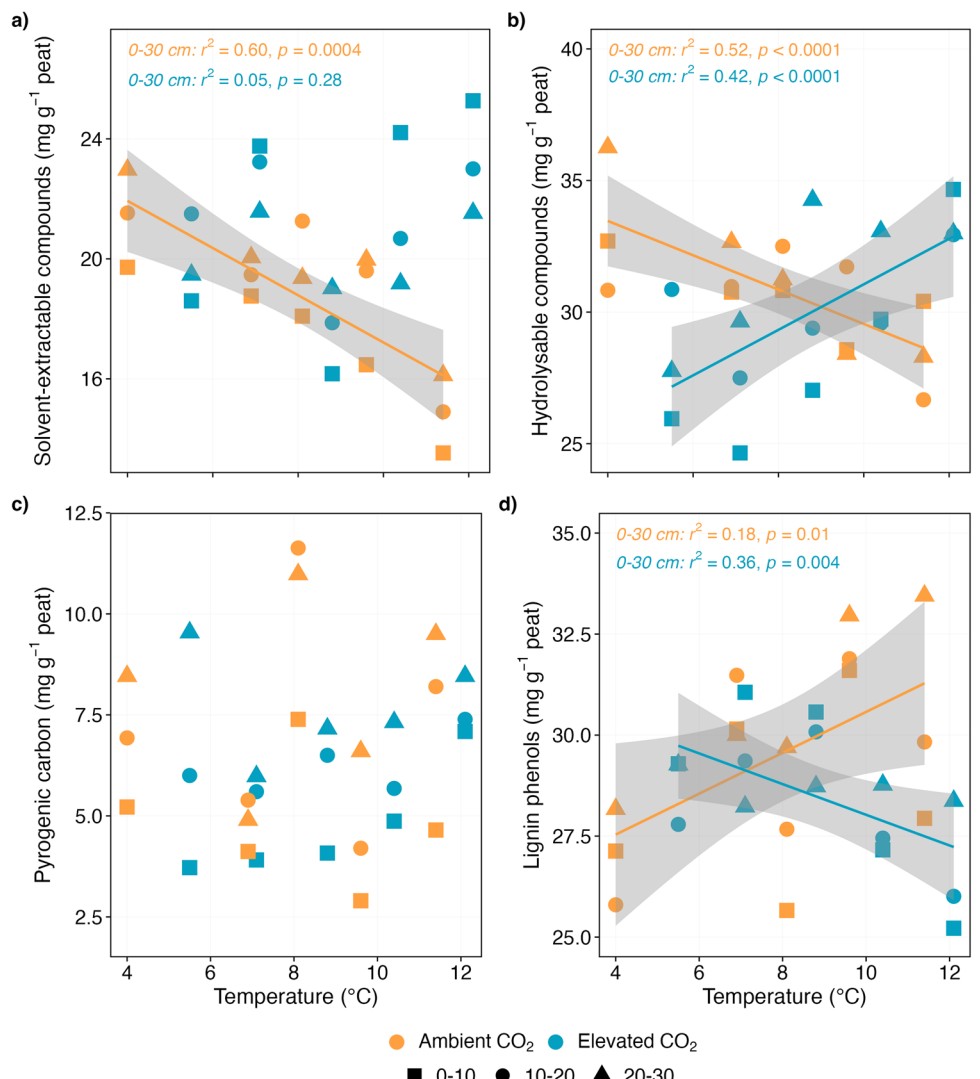

**Fig. 1 | The concentration of plant-, microorganism- and fire-derived SOC components.** Linear temperature response of the total sum of (**a**) solvent-extractable compounds (free lipids), (**b**) hydrolysable compounds (ester-bound lipids), (**c**) pyrogenic carbon and (**d**) lignin phenols in the surface peat (0–30 cm depth) following 4 years of warming and 2 years of elevated atmospheric $CO_2$ concentrations. The concentrations are plotted against average soil temperature measured at 0.3 m below the hollow surface from 2016 to 2018. Colours represent ambient (orange) or elevated $CO_2$ (blue) treatment ($n = 5$ per treatment). Symbols represent different sampling depths. Lines indicate significant treatment effects $p < 0.05$. Linear regression with 95% confidence intervals is shown in grey. The absence of a line and/or confidence intervals indicates no significant trend. Note that the concentrations in the deeper anaerobic peat (>40 cm depth) did not differ significantly with increasing temperatures (see Supplementary Fig. 5).

($-0.38 \pm 0.2$ mg g$^{-1}$ per °C warming; $r^2 = 0.36$, $p = 0.004$; Fig. 1d), raising possibilities of enhanced incorporation of other SOC compounds (i.e., hydrolysable biopolymers) into soil, leading to the decrease (dilution) of lignin phenols. Previous work at the study site demonstrated increased above- and belowground vascular plant biomass in response to warming and eCO$_2$ treatments[16,17,20]. As the plant litter input increases, so do the concentrations of SOC components in our results (Fig. 1; Fig. S2b), which suggest that warming and eCO$_2$ could stimulate SOC incorporation via increased plant litter inputs[17,42], whereas general impacts on C storage remain unclear. However, it is plausible that our observed increase in SOC components under eCO$_2$ reflect a transient adjustment period to the 2 years of eCO$_2$ treatment, whereby plant and microbial growth and traits, and decomposition rates have not yet reached a new equilibrium[43]. Alternatively, more rapid C cycling likely occurred within extant SOC components[41], although an increase in SOC turnover appears to have been offset by additional plant productivity[16,17]. Therefore, we posit that SOC will be partially preserved under warmer temperatures and

greater $CO_2$ concentration in the short-term, though this may change in the future.

The above responses in SOC components to warming (only) and warming plus eCO$_2$ were much greater in the surface peat (0–30 cm) than at deeper depths (>30 cm; deep peat) (Figs. 1, 3; Fig. S5-6). We observed a rapid turnover of SOC molecules in the surface peat (Figs. 1–3), a high degree of transformation that likely reflects a combination of altered plant C inputs and decomposition driven by low water tables during summer dry periods, compounded by warmer temperatures (Fig. 4). Warming was accompanied by a substantial water level drawdown during summer dry periods (30 cm below hollow surface in warmer plots)[17] in our study, and water table depth was correlated with SOC molecules (Fig. 4), suggesting that the warming and eCO$_2$ effect on SOC components was influenced by water level and downstream effects of peat oxygenation such as increased nutrient availability[44]. However, it is worth noting that previous studies have not yet observed a direct effect of warming on soil moisture in this experiment[17]. Four years into the treatment, deep anaerobic peat

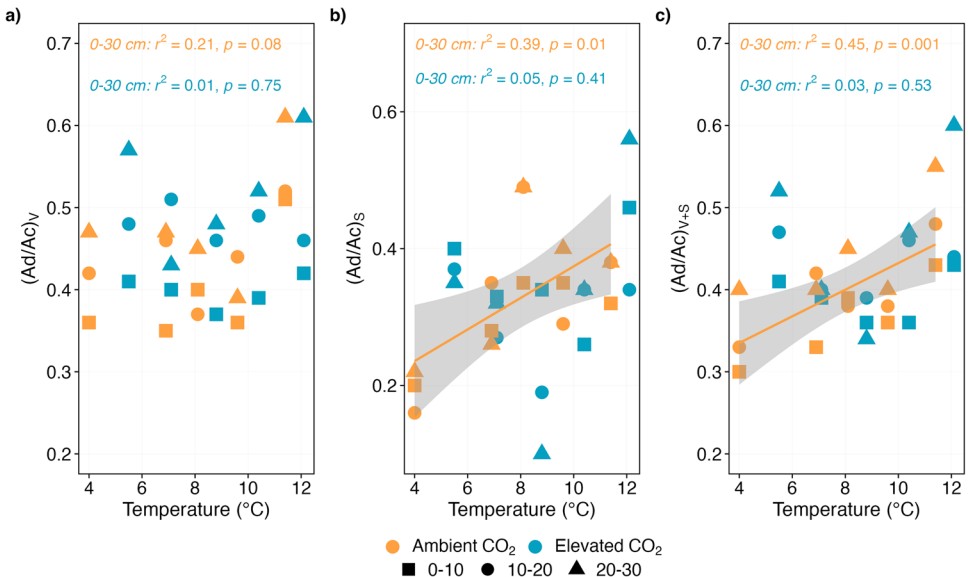

**Fig. 2 | Ratios of lignin oxidation (degradation) indices.** Linear temperature response of lignin oxidation proxy acid-to-aldehyde (Ad/Al) ratio for (**a**) vanillyl (V), (**b**) syringyl (S), and (**c**) vanillyl and syringyl phenols in the surface peat (0–30 cm depth) following 4 years of warming and 2 years of elevated atmospheric $CO_2$ concentrations. The ratios are plotted against average soil temperature measured at 0.3 m below the hollow surface from 2016 to 2018. Colours represent ambient (orange) or elevated $CO_2$ (blue) treatment ($n = 5$ per treatment). Symbols represent different sampling depths. Lines indicate significant treatment effects $p < 0.05$. Linear regression with 95% confidence intervals is shown in grey. The absence of a line and/or confidence intervals indicates no significant trend.

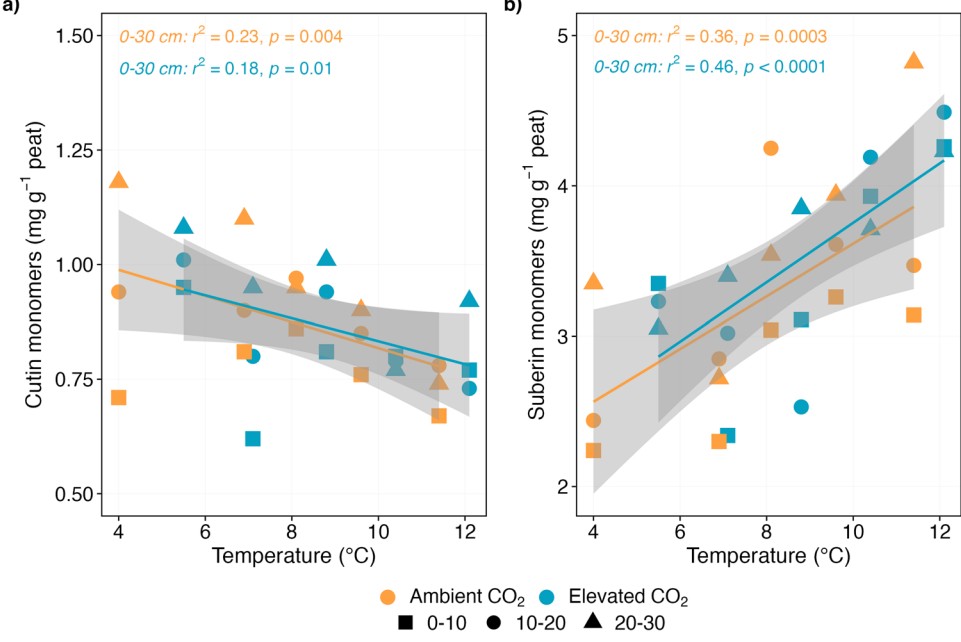

**Fig. 3 | The concentration of SOC components distinct to leaf/needle and root compartments.** Linear temperature response of the total sum of monomers distinct to (**a**) leaf/needle (cutin), and (**b**) root/bark (suberin) compartments in the surface peat (0–30 cm depth) following 4 years of warming and 2 years of elevated atmospheric $CO_2$ concentrations. The concentrations are plotted against average soil temperature measured at 0.3 m below the hollow surface from 2016 to 2018. Colours represent ambient (orange) or elevated $CO_2$ (blue) treatment ($n = 5$ per treatment). Symbols represent different sampling depths. Lines indicate significant treatment effects $p < 0.05$. Linear regression with 95% confidence intervals is shown in grey. The absence of a line and/or confidence intervals indicates no significant trend. Note that the concentrations in the deeper anaerobic peat (40–200 cm depth) did not differ significantly with increasing temperatures (see Supplementary Fig. 6). Biopolymers were selected according to their occurrence in the analysed leaves/needles, stems, and roots of the dominant plant species at SPRUCE (see Supplementary Table 2).

appears to be stable (Fig. S5-6) likely due to the slower decomposition that occurs under anoxic conditions[2,9] and low-quality C substrates in these layers[13,35]. It is also plausible that the deep anaerobic peat underwent extensive decomposition when it was part of the surface peat (1580-9200 BP)[45], creating peat that is less susceptible to the

changes that we observed in the surface layers. Together, our results suggest that warming and eCO₂ induce depth-dependent SOC responses, which should be carefully assessed to improve our mechanistic understanding and predictions of SOC dynamics under changing climate.

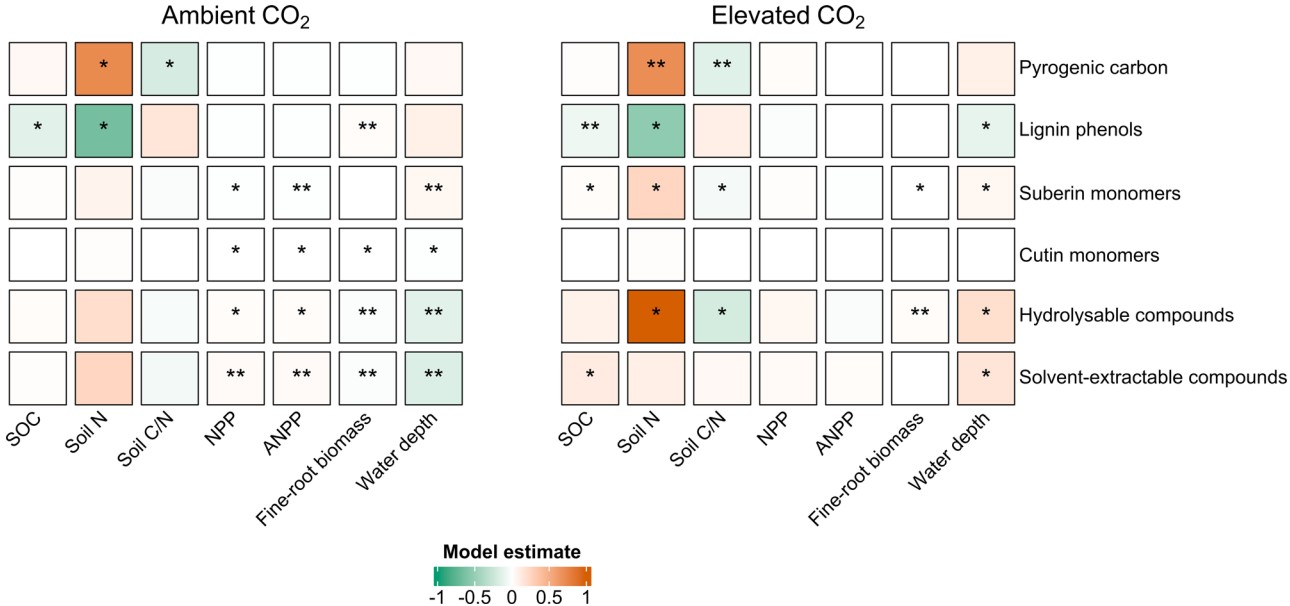

**Fig. 4 | Correlations (*r*) between SOC molecules and biogeochemical drivers.**
Pearson correlations between the total sum of solvent-extractable compounds
(free lipids), hydrolysable compounds (ester-bound lipids), cutin and suberin
monomers, pyrogenic carbon, and lignin phenols (mg g peat$^{-1}$) and biogeochemical
predictors soil organic carbon concentration (SOC; mg g peat$^{-1}$)[17,43], nitrogen con-
centration (soil N; mg g peat$^{-1}$)[43], carbon:nitrogen ratio (soil C/N; mg g peat$^{-1}$)[43], net
primary productivity (NPP; g C m$^{-2}$ year$^{-1}$)[17], aboveground net primary productivity
(ANPP; g C m$^{-2}$ year$^{-1}$)[17], fine-root biomass (g m$^{-2}$)[16], and maximum distance to the

water table (water depth; cm)[17,47] in the surface peat (0–30 cm depth) following
4 years of warming and 2 years of elevated atmospheric $CO_2$ concentrations. Cor-
rected significance at $p < 0.05$ is represented with *$p < 0.01$ is represented with **,
and $p < 0.001$ is represented with ***. The maximum distance (value) to the water
table (relative to the hollows) was assessed from 2016 to 2018. The maximum
values occurred in late summer (mid-August to mid-September). Additional
descriptions of biogeochemical predictors and any data transformations are pro-
vided in the Methods.

## Warming and eCO$_2$ enhanced the contribution of root-derived C inputs into SOC

To further delineate the contribution of plant litter inputs into SOC, we
assessed changes in the incorporation and degradation of biopolymers
distinct to either leaf/needle (cutin) or root/bark (suberin)
compartments[26–29] in the surface peat. The concentration of cutin
monomers decreased linearly with increasing temperatures under
both ambient $CO_2$ and e$CO_2$ (by -20% between the 0 and 9 °C treat-
ments, corresponding to -0.03 mg g$^{-1}$ per °C warming) ($r^2 = 0.23$,
$p = 0.004$ and $r^2 = 0.18$, $p = 0.01$ respectively; Fig. 3a; Fig. S3a). By
contrast, the concentration of suberin monomers increased linearly
with increasing temperatures in both ambient $CO_2$ and e$CO_2$ plots (by
~30% between the 0 and 9 °C treatments, corresponding to 0.19 mg g$^{-1}$
per °C warming) ($r^2 = 0.36$, $p = 0.0003$ and $r^2 = 0.46$, $p < 0.0001$
respectively; Fig. 3b; Fig. S3b). Thus, warming and e$CO_2$ led to the loss
of needle/leaf-derived C and gain of root-derived C in these soils
(Fig. 3; Fig. S3). Importantly, root-derived C (inferred from suberin
monomers) was positively correlated with SOC, fine root biomass and
water table depth (Fig. 4) and fine root biomass significantly increased
under warming and e$CO_2$ (Table S1)[16], implying increased root-derived
C inputs within the surface peat. In corroboration of our results, litter
manipulation experiments showed root-derived organic matter to be a
source of SOC with greater relative stability and longer turn-over
times[28], whereas leaf-derived C was found to be turned over more
rapidly in (mineral) soils[27,36,46]. Our study provides evidence that future
climate change may lead to increased retention of root-derived inputs
into SOC and a decline in leaf-derived C inputs in ecosystems such as
boreal forested peatlands. As warming is likely to be accompanied by
the decimation of peat-forming mosses and increases in fine-root
biomass in peatland ecosystems[16,47], it is likely that root-driven accrual
of SOC will represent an important mechanism for a continued C
sequestration in these ecosystems in the short-term. However, the
extent to which root-derived C can be incorporated and stabilized into
deeper peat layers will depend on the depth at which root inputs

increase. Thus, while we show that warming and e$CO_2$, in the short
term, enhanced incorporation of root-derived C into peat soils; it
remains to be seen what the longer-term fate of this root-derived
C will be.

## There is no inherently stable peat SOC

We observed a rapid turnover of complex SOC molecules (such as
lignin, cutin and suberin) and plant- and microbe-derived SOC com-
ponents under warming-only and warming plus e$CO_2$ (Figs. 1, 3; Fig.
S2), supporting the paradigm already observed in mineral soils that
there is no inherently stable SOC[24,25]. Complex SOC constituents such
as lignin have traditionally been considered to cycle more slowly (on a
multi-decadal time scale) relative to bulk SOC[23,32]. However, the
expectation that the enzymatic breakdown of complex SOC molecules
is more sensitive to temperature in comparison to simple molecules[34],
based in part on the kinetic theory, is not consistent with our results.
We show rapid turnover of SOC molecules irrespective of their origins
and potential decomposition rates (Fig. 1; Fig. S2). This high degree of
transformation likely reflects extensive decomposition driven by low
water tables[17] and was compounded by warm temperatures along with
increased labile organic matter[38] and plant inputs (Fig. 4). Our results
are consistent with previous studies reporting a decline in aromatic
compounds and soluble aromatics (indicative of lignin and pyrogenic
C) in the water-extractable organic matter with warming and e$CO_2$[38,39].
These results combined, show that microbial decomposition dynamics
in this peatland are controlled by a range of biotic and abiotic eco-
system properties and not only by the kinetics of individual enzyma-
tically catalysed reactions[24,25].

Unsurprisingly, molecules derived from historical pyrolysis (fire-
derived carbon or pyrogenic C; PyC) were unaffected by warming or
e$CO_2$ (Fig. 1c; Fig. S2c). We did not observe significant changes in PyC
concentrations under ambient $CO_2$ or e$CO_2$ ($p > 0.05$; Fig. 1c; Fig. S4a),
despite these peat soils containing PyC above global averages (~1.4% of
the SOC; Fig. S1)[48]. PyC did show an increasing trend with warming

under eCO$_2$ (Fig. 1c; $p > 0.05$) probably due to previous biomass burning and incorporation of residues before the SPRUCE experiment[49]. While these results are expected given that PyC is viewed as 'highly resistant organic matter'[23,30,32], recent evidence suggests that the increased stability of these SOC components does not guarantee long-term persistence[23]. To the best of our knowledge, no previous in-situ peatland manipulation studies have reported on the fate of PyC under changing climate drivers.

Collectively, our study provides evidence that rising temperatures and atmospheric CO$_2$ levels will have cascading effects on the turnover rates and controls of SOC compounds that dictate peatland C storage. Specifically, climate-induced warming and associated water table drawdown may increase oxygen availability and accelerate microbial decomposition of SOC. We also highlight a rapid turnover of complex SOC molecules in response to warming and eCO$_2$, supporting the paradigm that SOC stability depends on ecosystem properties[24,25]. Thus, SOC decomposition and turnover are predominantly determined by environmental and belowground plant-input constraints rather than temperature sensitivity of individual molecules[25]. While our results indicate that ecosystem responses at SPRUCE are largely driven by surface peat, it remains to be seen whether a system shift characterized by more rapid C cycling to deeper depths in the peat column will be created by extended water-table drawdown and associated changes in plant and microbial community composition[4,16,17,38] with further warming. Furthermore, we do not know whether the long-term SOC balance will be affected by plant–soil interactions or changes in soil microbial communities as they adapt to warmer temperatures[5]. Therefore, it is important to complement our results with long-term and time-resolved analyses. We also note that the results described here do not necessarily reflect the expected or observed responses for other peatland habitats. Nonetheless, the environmental gradients of SPRUCE provide us with model functions and parameters that can be applied to other similar peatland systems.

## Implications

Understanding soil C dynamics, including vulnerability to warming and eCO$_2$, is critical for climate prediction over the coming decades to centuries[5,7]. The unprecedented design of the SPRUCE whole-ecosystem warming experiment enabled us to quantify changes in key SOC molecules to reveal mechanisms of climate change responses in boreal peatland C storage and cycling. We demonstrate that all SOC components responded to climate change. Under scenarios of water table drawdown, the resulting aerobic conditions and higher summer temperatures could cause substantial SOC loss, at least in the short term. Conversely, under a scenario of combined warmer temperatures and elevated atmospheric CO$_2$ concentration, enhance SOC storage is possible if C loss is balanced or exceeded by increased primary productivity, particularly from enhanced root growth, but this could be a short-term response. Moreover, warming could still cause considerable and rapid decomposition of previously water-saturated peat, which could make C stocks in high-latitude peatlands more susceptible to losses in the future. Peatlands build C stocks over centuries[45], but rising temperatures and atmospheric CO$_2$ concentrations rapidly changed the equilibrium at SPRUCE within a 4-year timescale, highlighting the vulnerability of these C-rich ecosystems to global change.

## Methods
### Site description

The Spruce and Peatland Response Under Changing Environments (SPRUCE) experiment is located at the southern boundary of the boreal region on the Marcell Experimental Forest in northern Minnesota, USA (47°30'20.5"N, 93°27'12.6"W; http://mnspruce.ornl.gov/). Mean annual temperature and precipitation is 3.4 °C and 780 mm respectively[49]. The bog is ombrotrophic with peat depths of ~3 m

(which accumulated over the last 11,000 years), with a pH ranging from 4.1 at the surface to 5.1 at 2 m depth[45,49]. The bog is dominated by overstory trees, *Picea mariana* (black spruce) and *Larix laricina* (larch), and ericaceous shrubs, *Rhododendron groenlandicum* and *Chamaedaphne calyculata*, and bryophyte layer, primarily *Sphagnum* mosses. This ombrotrophic bog has a perched water table which fluctuates about 10–20 cm above the hollows after snowmelt, receding deeper later in the growing season[17,45].

### Experimental design and sample collection

The SPRUCE experimental design is described in detail by Hanson, et al. [33]. Briefly, the experiment consists of ten octagonal transparent open-top enclosures of 12 m diameter and 7 m height. Whole-ecosystem warming is maintained at a series of increasing temperatures (regression design) to five levels (+0, +2.25, +4.5, +6.75 and +9 °C) down to a depth of ~3 m using concentric arrays of electrical resistance heaters installed into peat and forced air warming. Peat warming was initiated between June and July 2014, following 3 months of a gradual treatment equilibration. Air warming was established in August 2015, thereby achieving whole ecosystem warming. In June 2016, eCO$_2$ treatment was introduced to the duplicate warming enclosures completing the planned experimental setup in this project. The eCO$_2$ treatment consists of elevating the local ambient atmospheric CO$_2$ concentration by +500 ppm (~900 ppm, with δ$^{13}$C–CO$_2$ isotope value of ~-54‰).

In August 2018, we randomly collected two soil cores from each of the plots in hollow microtopography where the surface of the hollow was defined as 0 cm. Surface samples (0–30 cm) were cut using a stainless-steel knife and extracted by hand, while a Russian peat corer was used to sample deeper peat (30–200 cm). Once collected, duplicate cores from the same plot were sectioned by depth (into 10 cm increments over 0 to 50 cm depth and 25 cm intervals from 50 to 200 cm), homogenized and combined to form a mixed sample. Dominant vascular plants and *Sphagnum* mosses were also randomly collected from each plot. Peat and plant samples were stored frozen at -20 °C immediately after sampling until further analyses. To avoid disturbance within the treatment plots, roots were collected from the dominant plants outside the enclosures and additionally picked from the soil manually. Peat sections and plant materials were later freeze-dried to constant weight. We sieved dried peat samples through a 5 mm mesh to remove larger litter fragments (2 mm sieve would have caused a loss of peat moss biasing the results). A subsample of the sieved peat and plant material was ground using a ball mill (MM400, Retsch, Haan, Germany) and then analysed for C concentration by an elemental analyser-isotope ratio mass spectrometer (EA-IRMS, Thermo Fisher Scientific, Bremen, Germany). Overall, SOC concentration increased with increasing depth from 44.0% at the surface to 52.5% at 2 m depth, but there was no effect of temperature or eCO$_2$ on the SOC concentration[43]. Additionally, investigations of SOC stock at the site so far have shown that the mean effect of warming on soil C stocks was indistinguishable from zero[17]. Our observations are not contradictory with this result since the specific compounds we followed represent 20% of the total SOC (see Fig. S1) and should be seen as tracers of SOC. Bulk density measurements were taken from soil cores taken from each of the experimental plots in 2013 and 2020. Soil cores 5.2 cm in diameter were carefully excavated using a Russian corer and soil bulk density was calculated using the freeze-dried weights of the volumetric slices. Bulk density was used to estimate the mass of SOC molecular components (stocks) in the top 30 cm (g m$^{-2}$). We calculated the mass of SOC molecular components by multiplying bulk density value (g soil cm$^{-3}$) by concentrations of individual SOC compounds (solvent-extractable compounds, hydrolysable biopolymers, lignin phenols and pyrogenic C) (g g peat$^{-1}$) from each peat depth and summarized the values for the mentioned depth intervals.

## Solvent-extractable compounds

Solvent-extractable lipids (alkanes, and alkanoic acids) in these soils have been previously reported[43]. Here, we extend our previous study by including alkanols, steroids and terpenoids in our analysis in order to assess the stability of total solvent-extractable lipids (herein, solvent-extractable compounds) with varied degradability. These soils were analysed using the protocol reported in Wiesenberg and Gocke[50]. Briefly, ~1 g of plant materials and ~2 g of milled peat was extracted using Soxhlet extraction with dichloromethane (DCM):methanol (MeOH) (93:7; v/v). The extracts were sequentially separated into low-polarity and acid (alkanoic acid) fractions using a KOH-coated (5%) silica gel column by eluting with DCM, and DCM:formic acid (99:1; v/v), consecutively. The low-polar fractions were further separated into aliphatic hydrocarbons (including alkanes, steranes and hopanes), polycyclic aromatic compounds, and hetero-compounds (including alcohols, sterols and steranes) using an activated silica gel column by eluting with hexane, hexane:DCM (1:1; v/v) and DCM:MeOH (93:7; v/v), consecutively. An aliquot of the alkanoic acid fraction was spiked with an internal standard (eicosanoic acid; $D_{39}C_{20}$) and derivatized to fatty acid methyl esters (FAMEs) prior to analysis using boron trifluoride:MeOH solution ($BF_3$:MeOH). Aliphatic hydrocarbons, polycyclic aromatic compounds, and hetero-compounds were spiked with tetracosane ($D_{50}C_{24}$), $D_{10}$-phenanthrene and octadecanol ($D_{37}C_{18}$), respectively, as an internal standard prior to analysis. Overall, analytical errors were typically <10% based on replicate analysis ($n = 6$). Given that alkanes, alkanoic acids, alkanols, steroids and terpenoids showed similar responses to both warming and eCO$_2$, we summed them to represent solvent-extractable compounds.

## Hydrolysable biopolymers

After pre-extraction of the solvent-extractable compounds, soil/plant residues were subjected to alkaline hydrolysis to extract ester-bound hydrolysable lipids (including cutin and suberin markers) as described by Mendez-Millan et al. [51]. Solvent-extracted residues (~0.5 g of plant and ~1 g of peat) were refluxed at 85 °C for 18 h with methanolic potassium hydroxide (KOH) (MeOH:water; 9:1, v/v; with 6% KOH). The extracts were filtered and acidified to pH 2.0 with 6 N hydrochloric acid (HCl). Compounds were recovered by liquid-liquid phase separation with DCM. The extracts were passed over sodium sulphate and dried under nitrogen gas. An aliquot of the extracts was spiked with an internal standard (eicosanoic acid; $C_{20}$) and derivatized with N,O-Bis-(trimethylsilyl)-acetamide (BSA; Merck, Germany) for 1 h at 80 °C. Samples were analysed in duplicates and analytical errors were typically <10% (we report mean values $n = 2$).

Biopolymers distinct to either leaf/needle (cutin) or root/bark (suberin) were selected according to their occurrence in the analysed leaves, stems, and roots of the dominant plant species at SPRUCE; *Picea mariana*, *Larix laricina*, *Rhododendron groenlandicum*, *Chamaedaphne calyculata*, and *Sphagnum* mosses (Table S2). Leaves, stems, and roots were characterised by different abundances and chain lengths of *n*-alcohols, *n*-carboxylic acid, α-hydroxy alkanoic acid, ω-hydroxy alkanoic acid, α,ω-alkanedioic acid and mid-chain-substituted hydroxy and epoxy alkanoic acids. The hexacosanoic acid (*n*-C$_{26:0}$), octacosanoic acid (*n*-C$_{28:0}$), dihydroxyoctadecanoic acid (9,10-diOH C$_{18:0}$) and dihydroxyoctadecenoic acid (x,18-diOH C$_{18:1}$) were used as markers for leaf/needle-derived C as they were either not detected or occurred in trace abundance in plant roots (Table S2). The ω-hydroxy alkanoic acids with a chain length of C$_{20}$, C$_{24}$ and C$_{26}$ and α,ω-alkanedioic acid with a chain length C$_{20}$, C$_{22}$ and C$_{24}$ (also α,ω-octadecenoic acids, C$_{18:1}$ diacid) were used as markers for root/bark-derived C as they were either not detected or occurred in trace abundance in leaf/needle (Table S2). These compounds correspond to previously suggested cutin and/or suberin monomers[27,51]. The unspecific monomers were treated as compounds derived from cutin or suberin, and they were summed to represent hydrolysable biopolymers (herein, hydrolysable compounds)[27,29].

## Lignin phenols

Bulk peat samples (milled) were subjected to copper (II) oxide (CuO) oxidation to extract lignin phenols[52,53]. Briefly, peat material equivalent to ~2 mg of organic carbon was mixed with 500 mg CuO, 50 mg of ammonium iron (II) sulphate hexahydrate and 20 mL of 2 M NaOH solution in Teflon-lined bombs. The bombs were then flushed with nitrogen for 10 min and heated (oxidized) in the microwave for 90 min at 150 °C. Once cooled, the oxidation products were spiked with ethylvanillin (3-ethoxy-4-hydroxy-benzaldehyde) and cinnamic acid (3-phenyl-2-propenoic acid) as the recovery and internal standards, respectively, and subsequently acidified to pH 2.0 using 6 M HCl. The oxidation products were then extracted from the aqueous phase with ethyl acetate, dried under nitrogen and redissolved in *p*-Anisic acid. An aliquot of the oxidation products was converted to trimethylsilyl derivatives with N,O-bis (trimethylsilyl) trifluoroacetamide/ tetramethylchlorosilane (BSTFA + TCMS; 99:1; Sigma-Aldrich, USA) by heating for 20 min at 60 °C. Two analytical replicates were measured for all samples (analytical errors <20%; we report mean values $n = 2$). Eight characteristic oxidation products were summarized to represent the concentration of lignin phenols, including vanillyl (vanillin, acetovanillone, vanillic acid), syringyl (syringaldehyde, acetosyringone, syringic acid) and cinnamyl (*p*-coumaric acid, ferulic acid) phenols (Fig. S4b). In addition, CuO oxidation released salicylic acid, salicylaldehyde, and piceol phenols (see Fig. S4b for individual fractional abundances). Lignin-specific oxidation products were used to calculate lignin oxidation (degradation) indices; acid-to-aldehyde ratios of vanillyl and syringyl phenols[32]. We note that while the CuO method is not interfered by any other organic component in peat soils, the method may significantly reduce the yield of phenolic units upon CuO oxidation[54,55].

The above SOC molecular compounds were quantified using internal standards on gas chromatography (GC) equipped with a multi-mode injector and a flame ionization detector (GC-FID, Agilent 7890B). Compound identification was performed on an Agilent 6890 N GC equipped with split/splitless injector coupled to an Agilent 5973 mass selective detector (MS) using external standard mixtures and by interpretation of the mass spectra. On the GC-FID/MS, the samples (1 μl) were injected in splitless mode. Both GC instruments were equipped with DB-5MS column (50 m x 0.2 mm × 0.33 μm), with helium as the carrier gas (1 ml min⁻¹). For solvent extractable compounds, details of the GC operating conditions are described elsewhere[50]. For hydrolysable biopolymers, the GC oven temperature was held at 50 °C for 4 min, increased to 150 °C at a rate of 4 °C min⁻¹, then increased to 320 °C at a rate of 3 °C min⁻¹, with final isothermal hold at 320 °C for 40 min. Quantification of the biopolymers was achieved by comparison with an external calibration with 16-hydroxyhexadecanoic acid. For lignin phenols, the temperature was held at 80 °C for 5 min, increased from 80 °C to 110 °C at a rate of 2 °C min⁻¹, then increased to 170 °C at a rate of 0.5 °C min⁻¹, and at a rate of 15 °C min⁻¹ to 320 °C, with final isothermal hold at 320 °C for 10 min.

## Pyrogenic carbon

Benzene polycarboxylic acids (BPCAs) were used as an approximation of pyrogenic carbon (PyC) and they were analysed following the protocol by Wiedemeier et al. [56]. Briefly, milled peat samples containing ~10 mg of organic carbon were digested with 65% nitric acid in the digestion tubes at 170 °C for 8 h. The extracts were filtered and passed over a cation exchange resin for further cleaning and subsequently freeze-dried. The dried extracts were redissolved in MeOH:water (1:1; v/v) and passed over a solid phase extraction cartridge (C$_{18}$ SPE tube, Supelco, U.S.A), then dried under nitrogen and redissolved in

deionized water. Compounds were quantified on a high-performance liquid chromatograph (HPLC) (Agilent 1290 Infinity HPLC system, Santa Clara, U.S.A.), equipped with an Agilent InfinityLab Poroshell 120 SB-C$_{18}$ column (100 mm × 4.6 mm × 2.7 mm), and measured using a photodiode array detector (DAD). Ortho-Phosphoric acid (Honeywell, U.S.A.) dissolved in water (pH 1.2) was used as mobile phase A and pure acetonitrile (Scharlau, Spain) as mobile phase B. Samples were analysed in duplicates and analytical errors were typically <10% (we report mean values $n$ = 2). All carboxyl functional groups (B3CA, B4CA, B5CA and B6CA) were summed to represent total concentration of PyC (see Fig. S4a for individual fractional abundances).

### Data analyses

General linear mixed-effect models were used to determine the effects of temperature, elevated $CO_2$ concentrations and peat depth on individual SOC compounds (solvent-extractable compounds, hydrolysable biopolymers, lignin phenols and pyrogenic C). We checked for bivariate relationships between all variables to ensure that a linear model was appropriate. Normality and homoscedasticity in the models were checked using residuals and Q-Q plots (none of the data set was adjusted to fit parametric assumptions). In all cases, linear regression models included plot as a random effect, and all other predictor variables (temperature, elevated $CO_2$ concentrations and peat depth) as fixed effects. For instances in which both temperature and $CO_2$ effects were significant, we evaluated separate regressions against temperature for ambient and elevated $CO_2$ treatments (Table S3). When significant differences among depths were detected, we conducted pairwise comparisons using Tukey's honest significant difference test. In the models, we used the actual temperature measured at -0.3 m below the hollows averaged over the period 2016 to 2018 (Table S1). We chose 0.3 m as a representative depth because the water table has fluctuated about 30 cm relative to hollow during the growing seasons[17].

Correlation heatmap (Pearson correlation) was calculated between individual SOC compounds (solvent-extractable compounds, hydrolysable biopolymers, lignin phenols and pyrogenic C) and possible predictors (SOC concentration, nitrogen concentration, carbon:nitrogen ratio, net primary productivity, aboveground net primary productivity, fine-root biomass, and maximum distance to the water table). Stepwise multiple linear regression with Akaike Information Criterion (AIC) as the model selection condition was used to investigate relationships between individual SOC compounds and the most important predictor variable(s). Only variables showing significant correlations with the investigated parameter ($p < 0.1$) were included as independent variables in the model. After the establishment of regressions, candidate models were carefully investigated for collinearity among the selected predictors and assumptions of normality by calculating variance inflation factors (a cut-off value of 1 when collinearity was considered not to exist) and graphically examining plots of residuals (log-transformation did not improve the overall distribution). In the multiple linear regression model, potential explanatory variables included: SOC concentration, nitrogen concentration, net primary productivity, fine-root biomass, and maximum distance to the water table (Fig. 4; Table S4).

Overall, the chosen level of significance was 5% ($p < 0.05$) in all statistical tests (unless stated otherwise). All data analyses were performed using the R v.4.2.0 (R Core Team, Vienna, Austria) using the RStudio interface v. 1.2.5033 (RStudio Team, PBC, Boston, MA).

## Data availability

Data sets used in this study are available in the online SPRUCE project archive at https://doi.org/10.25581/spruce.113/2202278 and the long-term storage in the U.S. Department of Energy's Environmental Systems Science Data Infrastructure for a Virtual Ecosystem (ESS-DIVE; https://ess-dive.lbl.gov).

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

## Acknowledgements

We thank E. Solly for field assistance, G. Santilli and Y. Brügger for laboratory assistance and F. Petibon and C. Zosso for inspiration. The University Research Priority Program Global Change and Biodiversity (URPP-GCB) at the University of Zurich for offering opportunities for helpful discussions and exchange. The Swiss National Science Foundation (SNF) grant, awarded to the DEEP C project (project 200021_172744) and the U.S. Department of Energy Office of Science, Office of Biological and Environmental Research Terrestrial Ecosystem Science Program, through support for the Oak Ridge National Laboratory which is managed by UT-Battelle, LLC, for the US DOE under Contract DE-AC05-00OR22725. N. Ofiti acknowledges funding from the Swiss National Science Foundation (Postdoc mobility grant P500PN_206708). J. Kostka and R. Wilson acknowledge funding from the Genomic Science Program, U.S. Department of Energy, Office of Biological and Environmental Research, under grant number DE-SC0023297. A. Malhotra acknowledges funding from the University of Zurich Stiftung für Wissenschaftliche Forschung (STWF-22-028), Swiss National Science Foundation (project 200021_215214) and COMPASS-FME, a multi-institutional project supported by the U.S. Department of Energy, Office of Science, Biological and Environmental Research as part of the Environmental System Science Program.

## Author contributions

M.W.I.S., N.O.E.O., A.M., G.L.B.W. and S.A. conceived the study; P.J.H. and C.M.I. designed and maintained the experiment; N.O.E.O., P.J.H., A.M., J.E.K., R.M.W. and C.M.I. carried out the field campaign; N.O.E.O. and G.L.B.W. carried out biomarker and geochemical laboratory analyses; R.M.W., P.J.H., N.O.E.O. and A.M. provided complementary data

(including SOC and nitrogen concentration, net primary productivity, fine-root biomass, and water table depth); N.O.E.O. analysed the data with help from A.M. and S.A., and wrote the original draft. All authors commented on the interpretation and presentation of the data and contributed to editing the original draft. Funding acquisition: P.J.H., M.W.I.S. and N.O.E.O.

## Competing interests

The authors declare no competing interests.
