## [Peer Review File · Nature Communications]

Climate warming and elevated CO₂ alter peatland soil carbon sources and stabilityREVIEWER COMMENTS

Reviewer #1 (Remarks to the Author):

This study used a warming x eCO₂ manipulation to measure changes in soil organic matter molecular composition in peat soils. They found that surface peat soil responded fairly quickly, with different compounds increasing or decreasing depending on if solely temperature changed or temperature combined with CO₂. They found roots contributed more to SOC, and leaves contribute less, with warming regardless of CO₂ level, and that deeper peat did not show changes in SOC compositions over the four-year period examined.

Overall I found this an interesting study that elucidated how the molecular composition of peat may shift under global change. We still lack general conceptual models describing soil organic matter molecular composition, and carefully collected datasets like this one spanning climate manipulations help reveal the drivers and constraints on SOC composition. I appreciate that the authors highlight that this was a short-term response – it is interesting how fast the peat responded but does not indicate the long-term dynamics (as they acknowledge). The finding relates to root vs shoot inputs was especially interesting.

I think this manuscript could be improved by constraining the focus to these interesting and novel findings and not making too many extrapolations to SOC stocks overall, which were not the focus of the manuscript. Throughout the paper and especially the last Discussion paragraph contained a good deal of text related to overall SOC storage, for which no data are provided in this paper. I feel the manuscript would be stronger if the authors stick more closely to what their data show. Changes in molecular composition can have implications for stocks and that is worth discussing, but themselves are interesting for the insights they provide into plant and microbial dynamics.

Another general comment is that at many points it was unclear when the authors were drawing conclusions from the data in this study vs results of others. It would be better if this was clearer throughout. I also found many points where the language was not as precise as I wanted it to be, making it hard to follow the logic (specific lines below). One last general comment is that I found the second paragraph of Introduction a bit muddled. It seems this is the main justification for the study but as written is too vague and doesn't clearly lay out what we know and remaining key unknowns.

Below I provide specific suggestions for areas to strengthen the text.

Line 34 – 'sequestration' implies action, uptake into a pool leading to long-term storage. I think you simply mean that peatland C is a large (and important) stock. I suggest you remove this word/rephrase this sentence.

Line 53 – 'by unknown mechanisms' is quite vague. What exactly do you mean here, that we know how the components of the system will respond individually but it's difficult to predict the net effect? Or something else? Please clarify.

Lines 63-67 – sentence is too long and rambling, it is difficult to understand what was done. This should be improved. Also, which of these components are assumed to be microbial-derived?

Lines 78-83 – Not arguing with the Arrhenius framework overall, but specifically for lignin, degradation proceeds primarily by the action of oxidative enzymes. I would think that oxygen availability is therefore the main bottleneck, as in reference 8. In wetlands/peatlands, might apparent temperature sensitivity with warming in fact be due to increased oxygenation under hotter, drier conditions? How will you be able to see this ‘simple kinetic effect’?

Line 91 – again, which are the microbial-derived compounds, what underpins that assumption?

Lines 99-104 – you seem to be posing several explanations for the increase in lignin – increased stabilization, more inputs, and ‘faster cycling labile C’. Please clarify the rationale for this last one (I do not understand), and if possible state whether you think there is more evidence for one or the other of these (or some combination) before concluding the lignin discussion.

Line 108 – ‘an overall loss of SOC molecules’ – given the previous discussion about lignin this does not seem accurate. Lignin decomp rates might be increasing (I’m convinced by Figure 2), but overall/on balance you are gaining lignin.

Line 126 – ‘enhanced incorporation of plant C inputs to SOC’ I’m confused here, lignin also comes from plants, do you mean plant C inputs other than lignin?

Lines 128-129 – enhanced C storage? But this manuscript addresses molecular composition, I don’t see any evidence for increased C stocks overall. Need to either attribute that finding to other papers or make a clearer link.

Lines 132-135 – You lost me with this long sentence, I can’t follow the logic.

Line 163 – I think by ‘vegetation composition’ you mean the molecular composition of vegetation? Or do you actually mean abundance and cover of specific species? Please clarify.

Lines 169-170 – your (great!) study is one site, one biome. I advise caution in over-extrapolating, perhaps say ‘in peatlands’ or ‘in boreal forest’

Figure 4 – Not sure I got much from this heat map. Did you consider instead providing a table describing best-fit models for each of the SOC components?

Line 185 – what do you mean by disproportionately lost? Lignin concentration increased with warming. I am also still left to wonder, which (non-measured) compounds increased with warming, if SOC overall did not change as stated in the Methods?

Line 210 – to show that they are occurring at the same rate, wouldn’t you need to discuss the slopes?

This might be something helpful to add to the figures that are cited.

Line 272 – there was no effect of the treatments on SOC concentration? This is surprising given the statement in lines 85-86 that all compounds are vulnerable to loss under warming. If all compounds decline, how is total SOC unchanged?

Line 318 – there are some known limitations of the CuO method for measuring lignin in soil (for instance, the two references below). Since in this study, you are assessing treatment effects on otherwise identical soils, I don't see a major issue as relative differences should be robust, but do think you should acknowledge that the method often underestimates lignin overall.

Hernes, P. J., Kaiser, K., Dyda, R. Y. & Cerli, C. Molecular trickery in soil organic matter: hidden lignin. *Environ. Sci. Technol.* 47, 9077–9085 (2013).

Klotzbücher, T., Kalbitz, K., Cerli, C., Hernes, P. J. & Kaiser, K. Gone or just out of sight? The apparent disappearance of aromatic litter components in soils. *SOIL* 2, 325–335 (2016).

Table S1 – why do the Elevated CO₂ plots tend to have higher soil temperatures within the same nominal warming class?

Figure S7 – it's nice to see the soil bulk density data since compound abundances are presented as stocks. Please briefly address how and when bulk density was measured in the Methods

Data access – when submitting a revised version, will you have submitted datasets and thus can provide a DOI? This would be ideal. If not, can you describe how interested parties can find your datasets within the named repositories?

Reviewer #2 (Remarks to the Author):

This manuscript presents results on the effect of warming and elevated atmospheric CO₂ concentrations in experimental enclosures in a peatland in Minnesota, over a four-year period. The study provides strong evidence of changes in the chemical composition of the upper layers of the peat associated with both warming and elevated CO₂; the former leads to decreases in selected organic components, associated with enhanced decomposition, while the doubling of CO₂ concentration increased components associated with plant growth, such as roots and leaves/needles. The manuscript is generally well written and structured and presents evidence of possible changes in organic soil components resulting from 'climate change'. The SPRUCE study was designed to determine the effects of warming and elevated CO₂ concentration, with a clear recognition that the experimental manipulations (warming up to 9°C and CO₂ concentration increased by about 300 ppm) do not represent 'real' climate change but give some indication of the expected changes in direction, rather than in magnitude. SPRUCE has been very successful in its operation and the wide range of expertises that have been assembled at the site.

However, I do think that the results, although impressive, need to be presented with some caution: the environmental changes occurred more or less overnight, the range of change is beyond what might be expected to occur (though 2023 may seem to be a portent of rapid changes...), the fall in water table may be moderated by lateral flow in open systems and bogs have many adaptive mechanisms which has allowed them to survive changes in the past. Thus, I suggest the authors tone down some of the statements made, less dramatic and more cautionary in whether the changes shown at SPRUCE will occur in the vast area of northern bogs. The cautionary tone appears in lines 216-219 and it might be applied in the text. Particularly important is the recognition that increases in some SOC components (suberin) are linked to below-ground plant activities (roots), which are expected to have a slower turnover time, with vegetation responding positively to a warmer and drier peat soil.

I provide some specific comments, by line #:

1 'rapidly' seems a bit strong given the heavy jolt the system was exposed to and the short duration of the study (4 years temperature change, 2 years CO₂ change).

26 I prefer 'may' rather than 'will', and it seems that the lowering of the water table may be a significant contributor to the observe changes.

65 What would be the timescale of turnover of these SOC components in northern peat soils: the components are placed in three cycling groupings.

72 The planned increase of 500 ppm CO₂ seems not to have been achieved, from Table S1, more like 300 pp increase.

93 The title implies speed of change and one way to indicate that is to express the rate of change in the SOC components not just as a function of the change in temperature but over the four years of warming. Fig. 1a shows a regression in which there appears to be a decline in 'solvent-extractable compounds' from about 23 mg/g at ambient temperature (4oC), to about 16 mg/g at 11.4oC (9oC warming treatment): a decline of about 7 mg/g or about 30% of original content over 4 years. Thus, at the warmest treatment, that would amount to an average loss of about 8% per year (over the four years) and at a more 'realistic' warming treatment of 2.25oC (resulting in a soil temperature of about 5.5oC by eyeballing Fig. 1a) of about 1.5 mg/g or 6% of the original over four years, or 1.5% per year. Of course, the changes may be fast at the outset and slow down.

I am not sure whether the above is correct (happy to be corrected), but I wonder how the figure of 7.6 mg/g decrease in solvent-extractable compounds per oC increase in temperature was derived. If the temperature rose by 7oC, then all the original component would be gone. Please correct me if I have this wrong.

112. It seems that the lowering of the water table could be a significant response to the warming and elevated CO₂, in terms of the increase in the aerobic zone, possible increase in available nutrients through accelerated decomposition and potential increase in rooting zone. The only metric cited appears

to be the maximum depth of the water table over three years, which occurs in August-September. The maximum depth increases from 17 cm to 44 cm (Table S1), a lowering of 27 cm. Is there a more useful metric to indicate the drying of the peat, from the available data? There are few warming experiments of the scale of SPRUCE, but quite a few studies have examined the effect of drainage on peat chemistry and vegetation patterns.

160. Reference might be made to Table S1 in which fine root biomass is presented and shows a six-fold increase from the 'control' to the +9 plots (55 and 327 g/m², respectively) and there seems to be a general increase among the plots.

The data are presented primarily as concentrations per gram of peat, though in Fig. S2 an estimate is made of the mass of components in the top 30 cm. Is there any evidence of the loss of SOC (e.g. g/m²) from the four years of treatments or over a longer period (e.g. the 2012 and 2020 bulk density data in the Supplementary Information)?

The components measured (Fig. 1&3) contribute a total of about 95 mg per gram of peat in the ambient plots. How might the remaining 90% of the peat react to the changes?

Fig. 1. I assume the regression is driven through the data for all three depths, and there is little differentiation among the 0-10 cm, 10-20 cm and 20-30 cm depths.

Tim Moore

The present manuscript was previously submitted to *Nature Communications* (NCOMMS-23-30015-T) and was assessed by two reviewers. We appreciate the extensive comments provided by the reviewers. We have substantially revised the manuscript based on the feedback we received; major changes are as follows:

- 1) The reviewers suggested constraining the focus of the manuscript on changes in SOC molecular composition and reducing broad extrapolations to SOC stocks. Furthermore, they suggested introducing caveats around some statements and discuss uncertainties and limitations. We have removed the more speculative aspects of our interpretation and have ensured that our work cannot be misconstrued as a study of soil C stocks. We have also reduced focus on broad extrapolations and toned down the strength of language throughout.
- 2) We have also clarified that the aims of the study were to investigate whether warming and elevated atmospheric CO₂ concentration (eCO₂) affects SOC stability via changes in molecular compounds comprising SOC and to identify which SOC molecules are susceptible to climate driven changes. We have clarified that we are talking about SOC molecular components that are best suited to infer mechanisms and not SOC stocks. These corrections have not altered our conclusions but have improved the coherence of the results.

We are confident that the revisions have addressed all of the reviewers' concerns and we give our detailed responses to each point below, along with references to the revised manuscript text when applicable.

Reviewers' comments:

Reviewer #1 (Remarks to the Author):

This study used a warming x eCO₂ manipulation to measure changes in soil organic matter molecular composition in peat soils. They found that surface peat soil responded fairly quickly, with different compounds increasing or decreasing depending on if solely temperature changed or temperature combined with CO₂. They found roots contributed more to SOC, and leaves contribute less, with warming regardless of CO₂ level, and that deeper peat did not show changes in SOC compositions over the four-year period examined.

Overall, I found this an interesting study that elucidated how the molecular composition of peat may shift under global change. We still lack general conceptual models describing soil organic matter molecular composition, and carefully collected datasets like this one spanning climate manipulations help reveal the drivers and constraints on SOC composition. I appreciate that the authors highlight that this was a short-term response – it is interesting how fast the peat responded but does not indicate the long-term dynamics (as they acknowledge). The finding relates to root vs shoot inputs was especially interesting.

I think this manuscript could be improved by constraining the focus to these interesting and novel findings and not making too many extrapolations to SOC stocks overall, which were not the focus of the manuscript. Throughout the paper and especially the last **Discussion paragraph** contained a good deal of text related to overall SOC storage, for which no data are provided in this paper. I feel the manuscript would be stronger if the authors stick more closely to what their data show. Changes in

molecular composition can have **implications for stocks and that is worth discussing**, but themselves are interesting for the insights they provide into plant and microbial dynamics.

Response: We thank the reviewer for the constructive and insightful comments.

There is compelling evidence from observational studies across climate gradients, and field warming experiments, that more rapid decomposition is not synonymous with reductions in total soil C stocks, at least in the short-term (<1 to ~10 years) (Lu et al., 2013; Bradford et al., 2016; Melillo et al., 2017; Song et al., 2019). The lack of direct evidence for reductions in soil C stocks, which is the case in our experiment (Hanson et al., 2020; Ofiti et al., 2022) makes it difficult to quantify the effects of warming on total soil C. Thus, this encouraged the use of molecular compounds comprising SOC (indirect measurements), to understand warming effects, and implications for overall SOC storage. As pointed out by the reviewer, “changes in molecular composition can have implications for stocks and that is worth discussing”, we therefore aimed to investigate the effects of warming and elevated atmospheric CO₂ concentration on peatland SOC molecular composition and use the observed changes in molecular composition to infer SOC sources (insights on plant and microbial dynamics), stability, and implications for storage. Given that several peatland climate change experiments exist (e.g., Bragazza et al., 2013; Walker et al., 2016; Liu et al., 2022), and biomarkers are an established tool to understand ecosystem processes (Feng et al., 2008; Pisani et al., 2015; Pold et al., 2016; Cheng et al., 2017; Bailey et al., 2018; Jia et al., 2019; Zosso et al., 2023), investigations on peat SOC molecular composition responses to changing environment are scarce. This makes our study unique as our biomarker technique is applied in SPRUCE’s whole-ecosystem warming setting. We are also unaware of any large-scale studies that has measured the changes we demonstrate in our study.

We have revised the manuscript to focus solely on SOC molecular components that are best suited to infer mechanisms and not SOC stocks and we acknowledge discussing the implications these changes have on C storage.

Another general comment is that at many points it was unclear when the authors were drawing conclusions from the data in this study vs results of others. It would be better if this was clearer throughout. I also found many points where the language was not as precise as I wanted it to be, making it hard to follow the logic (specific lines below). One last general comment is that I found the second paragraph of Introduction a bit muddled. It seems this is the main justification for the study but as written is too vague and doesn’t clearly lay out what we know and remaining key unknowns.

Response: We have removed speculative discussions and streamlined the results and discussion sections of the manuscript, by focusing on discussing the meanings indicated by our measurements, for example in line 108-112, 115-120, 142-148, 160-165. We now believe that the novelties of our study are strengthened in the revised manuscript. We have also clarified wording where it was unclear what was our study vs previous work.

We have revised the second paragraph of the Introduction to focus solely on what we know and remaining key unknowns in the stability of peat SOC, and their responses to climate change factors. It now reads:

“For projections of peatland C balance under future climate scenarios, a key question is whether a portion of SOC that is currently stable may become more accessible to microbial decomposition^{2,4,7}, via direct effects of rising temperature, indirect effects of altered water table^{9,14,15}, or due to changes in plant production and thus litter inputs to SOC^{11,16,17}. Soil warming is occurring more rapidly at high compared

to low latitudes^{18,19}, which can lead to lower water-tables and increases in peat aeration^{9,15}, with alterations in plant biomass production and in the quality of litter inputs^{10,16,20}. Such a change in peat aeration and plant litter (and root exudate) inputs can impact the microbially-mediated turnover of belowground soil carbon because soil water content and litter chemistry regulate the composition and activity of the microbial community in peat soils⁸⁻¹¹. Differences in litter chemistry can also affect the sensitivity of organic matter decomposition to soil temperature, with further effects on C cycling^{10,21}. Consequently, if these alterations in organic matter source persist over extended periods, they are likely to stimulate the mineralisation of currently stable C²². In short, climate-driven changes in water-table and plant litter inputs are likely to modify C storage^{5,6}, but the underlying mechanisms and their interactions, particularly those resulting in shifts in organic matter inputs and composition are still unknown in peatlands⁷. The role of such plant-soil interactions in regulating future SOC storage merits attention because a much larger proportion of SOC may be susceptible to climate-mediated losses than previously assumed^{7,19}.” (Line 46-62).

Below I provide specific suggestions for areas to strengthen the text.

Comment 1: Line 34 – ‘sequestration’ implies action, uptake into a pool leading to long-term storage. I think you simply mean that peatland C is a large (and important) stock. I suggest you remove this word/rephrase this sentence.

Response: We disagree with the reviewer. Radiocarbon data in particular show that peatlands than nearly any other ecosystems have been active in sequestering carbon for over 10,000 years (Hobbie et al., 2016; Griffiths et al., 2017; Wilson et al., 2021a). We have now added the statement that “The capacity of peatlands to store more than one-third of the global SOC pool under waterlogged conditions^{1,2} has been touted as a powerful natural form of C sequestration³.”(Line 33-34).

Comment 2: Line 53 – ‘by unknown mechanisms’ is quite vague. What exactly do you mean here, that we know how the components of the system will respond individually but it’s difficult to predict the net effect? Or something else? Please clarify.

Response: We agree and have removed this statement. We have added a paragraph to explain that climate-driven changes in water-table and plant litter inputs have the potential to modify C storage, but the underlying mechanisms are still unknown in peatlands, particularly around the sources and stability of organic matter (Chaudhary et al., 2020; Wilson et al., 2022). We have revised this sentence it now reads “In short, climate-driven changes in water-table and plant litter inputs are likely to modify C storage^{5,6}, but the underlying mechanisms and their interactions, particularly those resulting in shifts in organic matter inputs and composition are still unknown in peatlands⁷.” (Line 57-60).

Comment 3: Lines 63-67 – sentence is too long and rambling, it is difficult to understand what was done. This should be improved. Also, which of these components are assumed to be microbial-derived?

Response: We have now changed the sentence completely, breaking it into three parts and specifying which compounds are plant- and microorganism-derived. This sentence it now reads:

“Specifically, we targeted SOC compounds with distinct turnover times. Firstly, we targeted solvent-extractable compounds (alkanoic acids, alkanols, alkanes, steroids, and terpenoids, derived from plant- and microbial-material²⁶) that are thought to turn over predominantly on a timescale of bulk SOC or faster^{24,27}. Secondly, we targeted SOC components from different plant origins, hydrolysable biopolymers distinct to either leaf/needle (cutin) or root/bark (suberin) compartments (and presumably

slowly cycling)^{26–29}. Lastly, we targeted lignin and pyrogenic carbon (PyC) representing the more slowly cycling C^{23,30–32}.” (Line 67-73).

We do have components that are *only* microbial derived. Among the myriad of SOC constituents, solvent-extractable compounds, including alkanolic acids, alkanols, alkanes, steroids, and terpenoids, may yield information on the relative contribution of plants and microbes to SOC as they are derived from plant tissues and microorganisms (Kögel-Knabner, 2002; Otto et al., 2005; Angst et al., 2016). Plant-derived organic matter is characterized by long-chain alkanolic acids, alkanols, alkanes, diacids, and terpenoids whereas compounds such as *iso*- and *anteiso*-branched alkanolic acids are only derived from microorganisms (Harwood and Russell, 1984; Kramer and Gleixner, 2006; Wiesenberg et al., 2008). Additionally, with the decomposition of plant materials, soil microorganisms convert available carbon (including plant-derived organic matter) into microbial necromass or microbial processed compounds along with their own biomass (Miltner et al., 2012; Cotrufo et al., 2015; Kallenbach et al., 2016). We thus refer to solvent-extractable compounds as plant- and microorganism-derived as compounds such as *iso*- and *anteiso*-branched alkanolic acids and the microbial processed compounds contributed to the solvent-extractable portion of SOC.

At best, we can get at plant-only through the other category of hydrolysable polymers where we target biopolymers distinct to either leaf/needle (cutin) or root/bark (suberin) compartments. As some of the specific biomarker terms may not be familiar to the broad readership of *Nature Communications*, we have tried to keep the text accessible.

Comment 4: Lines 78-83 – Not arguing with the Arrhenius framework overall, but specifically for lignin, degradation proceeds primarily by the action of oxidative enzymes. I would think that oxygen availability is therefore the main bottleneck, as in reference 8. In wetlands/peatlands, might apparent temperature sensitivity with warming in fact be due to increased oxygenation under hotter, drier conditions? How will you be able to see this ‘simple kinetic effect’?

Response: Good point! Sphagnum peatlands have been suggested to possess a particularly effective mechanism for restricting enzymic decomposition under the “enzyme latch” hypothesis, whereby oxygen constraints associated with waterlogging suppress the activity of phenol oxidase enzymes (Freeman et al., 2001; Ise et al., 2008; Fenner and Freeman, 2011). This, in turn, allows an accumulation of enzyme-inhibiting phenolic compounds that impede nutrient cycling and decomposition. However, rather than just waterlogging, multiple constraints including temperature, pH, and biotic factors such as plant and microbial community composition and functioning as well as the chemical characteristics of the peat itself control SOC stability (Wang et al., 2015). As warming is typically accompanied by lower water-table and increased oxygenation, we agree that lower water-table and increased oxygenation have an increasing influence on the apparent temperature sensitivity of organic matter decomposition. We have modified this sentence it now reads:

“From the perspective of long-term peatland C sequestration, we are particularly interested in the response of polymeric SOC components because kinetic theory predicts that decomposition and turnover of complex SOC molecules (greater chemical stability) should be more sensitive to temperature changes than simple molecules³⁴. In peatlands, warming is typically accompanied by a lower water-table coupled with peat aeration, which can further impact the apparent temperature sensitivity of organic matter decomposition^{8,9,14,15}. Consequently, if temperature and oxygenation effects predominate over other biotic and abiotic factors that influence decomposition dynamics^{9,25,34}, we hypothesize that complex SOC molecules such as lignin, and PyC that have been proposed to turn over

much slower than bulk C in soils due to their polyaromatic chemical structure^{23,30,35}, would degrade faster.” (Line 81-90).

Comment 5: Line 91 – again, which are the microbial-derived compounds, what underpins that assumption?

Response: See previous comment 3 (Line 67-73).

Comment 6: Lines 99-104 – you seem to be posing several explanations for the increase in lignin – increased stabilization, more inputs, and ‘faster cycling labile C’. Please clarify the rationale for this last one (I do not understand), and if possible state whether you think there is more evidence for one or the other of these (or some combination) before concluding the lignin discussion.

Response: The slowly cycling compounds (e.g., lignin) can increase in relative abundance, if more labile compounds decrease in concentration, “dilution effect”. At SPRUCE, warming has increased the availability of labile sugars (and protein) (Wilson et al., 2021b). We thus posit that the relative increase in lignin phenols may be due to preferential decomposition of labile sugars and dilution of slowly cycling lignin phenols by an enhanced plant input. We have added a paragraph explaining why the increase in lignin implies increased stabilization, more inputs, and faster cycling labile C. This sentence now reads:

“Indeed, it is plausible that more lignin phenols entered soils via increased root litter production or that a shift in plant species composition toward shrubs richer in these compounds^{16,17} may explain this increase. It is also possible that this apparent increase in lignin phenols may be due to accelerated decomposition of more rapidly cycled labile C and dilution of slowly cycling lignin phenols, given that warming has increased the availability of labile sugars (and protein) in this experiment³⁸.” (Line 107-112).

We have also concluded on what we believe is the reason for increase in lignin as per the reviewer’s suggestion. We have added the following statement “The observed relative increase in lignin phenols (Fig.1d; Fig. S1) thus reflects a combination of increased organic matter inputs via plant litter and root exudates, enhanced degradation of more labile SOC molecules induced by higher temperatures^{16,38} and a better resilience of these molecules to warming as compared to the more labile ones.” (Line 115-118).

Comment 7: Line 108 – ‘an overall loss of SOC molecules’ – given the previous discussion about lignin this does not seem accurate. Lignin decomp rates might be increasing (I’m convinced by Figure 2), but overall/on balance you are gaining lignin.

Response: We changed the formulation and hope we can now avoid any confusion. “Overall, the above findings indicate that warming stimulated a loss of SOC molecules, irrespective of their origins and potential decomposition rates.” (Line 118-120).

Comment 8: Line 126 – ‘enhanced incorporation of plant C inputs to SOC’ I’m confused here, lignin also comes from plants, do you mean plant C inputs other than lignin?

Response: Apologies for this confusion. We have modified this sentence it now reads “Surprisingly, the concentration of lignin phenols decreased linearly with increasing temperatures ($-0.38 \pm 0.2 \text{ mg g}^{-1} \text{ per } ^\circ\text{C}$ warming; $r^2 = 0.36$, $p = 0.004$; Fig. 1d), raising possibilities of enhanced incorporation of other SOC

compounds (i.e., hydrolysable biopolymers) into soil, leading to the decrease (dilution) of lignin phenols.” (Line 140-143).

Comment 9: Lines 128-129 – enhanced C storage? But this manuscript addresses molecular composition, I don’t see any evidence for increased C stocks overall. Need to either attribute that finding to other papers or make a clearer link.

Response: Our study investigated the consequences of warming and elevated atmospheric CO₂ concentration on peatland SOC components and our results provide key evidence to suggest that C losses in soil are increasing via enhanced degradation, and C inputs are increasing because of increase of plant biomass, but the net change in SOC is not clear. We asked which molecular compounds comprising SOC were vulnerable to shifts in climate drivers and whether changes in decomposition rates and plant-derived inputs would alter the overall stability of SOC. As mentioned previously in our response to the reviewer’s general comment, due to the lack of direct evidence for reductions in soil C stocks, which is the case in our experiment (Hanson et al., 2020; Ofiti et al., 2022) and a synthesis of field data (Lu et al., 2013; Song et al., 2019), we opted for a different approach. We partitioned bulk soil C into classes of molecular compounds as they can provide unparalleled insight into SOC pools that are differently vulnerable to changing climate.

Based on reviewer comments, we have revised the manuscript and made it clearer that we are talking about SOC molecular components that are best suited to infer mechanisms and not SOC stocks. This sentence now reads:

“Previous work at the study site demonstrated increased above- and belowground vascular plant biomass in response to warming treatments^{16,17,20}. As the plant litter input increases, so do the concentrations of SOC components in our results (Fig. 1; Fig. S2b), which suggest that warming and eCO₂ could stimulate SOC incorporation via increased plant litter inputs^{17,42}, whereas general impacts on C storage remain unclear.” (Line 143-148).

Comment 10: Lines 132-135 – You lost me with this long sentence, I can’t follow the logic.

Response: We have modified this sentence it now reads “Alternatively, more rapid C cycling likely occurred within extant SOC components⁴¹, although an increase in SOC turnover appears to have been offset by additional plant productivity^{16,17}.” (Line 150-152).

Comment 11: Line 163 – I think by ‘vegetation composition’ you mean the molecular composition of vegetation? Or do you actually mean abundance and cover of specific species? Please clarify.

Response: Thank you for pointing this out. We have added the following statement to acknowledge the fact that neither warming nor CO₂ led to changes in molecular composition of the overlying plant biomass. “Thus, warming and eCO₂ led to the loss of needle/leaf-derived C and gain of root-derived C in these soils (Fig. 3; Fig. S3). These losses are not related to changes in the chemical quality of the vegetation since neither warming nor eCO₂ led to changes of the overlying plant quality^{16,43}.” (Line 183-186).

Comment 12: Lines 169-170 – your (great!) study is one site, one biome. I advise caution in over-extrapolating, perhaps say ‘in peatlands’ or ‘in boreal forest’

Response: Good point! We have toned down the extrapolation as you pointed out. This sentence now reads “Our study provides evidence that future climate change may lead to increased retention of root-derived inputs into SOC and a decline in leaf-derived C inputs in ecosystems such as boreal forested peatlands.” (Line 191-193).

Comment 13: Figure 4 – Not sure I got much from this heat map. Did you consider instead providing a table describing best-fit models for each of the SOC components?

Response: We used stepwise multiple linear regression to investigate relationships between individual SOC compounds and the most important predictor variable(s). The most important predictor variables were those that showed significant correlations with the investigated SOC components; this is why we decided to report correlations and not best-fit models. We would prefer to hold on to the presentation of heat map (Figure 4), as we think this helps to focus on the biogeochemical drivers of SOC molecules. However, we do agree that the best-fit models should be shown in the manuscript. In the revised manuscript we provide a table describing best-fit models for each of the SOC components (Supplementary Table 4).

Supplementary Table S3. Slopes of linear regression model when predicting solvent-extractable compounds (free lipids), hydrolysable compounds (ester-bound lipids), monomers distinct to leaf/needle (cutin), and root/bark (suberin) compartment, pyrogenic carbon, and lignin phenols using soil temperature. Each reported slope is from a single bivariate linear regression run either by ambient or elevated CO₂ treatment in the surface peat (0-30 cm depth) following 4 years of warming and 2 years of elevated atmospheric CO₂ concentrations.

Response variable	CO₂ treatment	Intercept	Slope estimate	Std. Error	P value
Solvent-extractable compounds	Ambient	25.08	-0.79	0.17	0.0004
	Elevated	18.21	0.32	0.29	0.28
Hydrolysable compounds	Ambient	36.07	-0.65	0.17	0.002
	Elevated	22.41	0.86	0.26	0.006
Cutin monomers	Ambient	1.10	-0.03	0.01	0.046
	Elevated	1.08	-0.02	0.01	0.07
Suberin monomers	Ambient	1.86	0.18	0.06	0.01
	Elevated	1.78	0.20	0.05	0.002
Pyrogenic carbon	Ambient	6.52	0.03	0.28	0.922
	Elevated	4.47	0.20	0.18	0.304
Lignin phenols	Ambient	25.51	0.51	0.22	0.04
	Elevated	31.83	-0.38	0.15	0.026

Supplementary Table S4. Optimal mixed-effects linear regression models to explain variation in solvent-extractable compounds (free lipids), hydrolysable compounds (ester-bound lipids), monomers distinct to leaf/needle (cutin), and root/bark (suberin) compartment, pyrogenic carbon, and lignin phenols. This table includes all possible predictors for SOC compound in either ambient or elevated CO₂ plots in the surface peat (0-30 cm depth) following 4 years of warming and 2 years of elevated atmospheric CO₂ concentrations. In the table, SOC denotes soil organic carbon concentration, soil N denotes nitrogen concentration, NPP denotes net primary productivity, and ANPP denotes aboveground net primary productivity. Models are reported for significance criteria with $\alpha < 0.05$. Values in parentheses are standard errors. Significant effects are marked with * for $p < 0.05$, ** for $P < 0.01$, *** for $p < 0.001$, and **** for $p < 0.0001$.

Response variable	CO ₂ treatment	Significance criterion	Predictor	Coefficient	Predictor	Coefficient	Predictor	Coefficient	Predictor	Coefficient	Adjusted R ²
Solvent-extractable compounds	Ambient	$p < 0.01$	Intercept	16.72*** (2.64)	ANPP	0.02* (0.01)	Fine-root biomass	-0.01 (0.005)	--	--	0.67
	Elevated	$p < 0.01$	Intercept	-23.31 (15.80)	SOC	0.10* (0.03)	Water table depth	0.19* (0.08)	Soil N	-0.55 (0.37)	0.52
Hydrolysable compounds	Ambient	$p < 0.05$	Intercept	37.90*** (3.26)	Fine-root biomass	-0.02** (0.004)	Soil N	-0.32 (0.24)	--	--	0.51
	Elevated	$p < 0.05$	Intercept	-12.42 (16.89)	Fine-root biomass	0.01* (0.005)	SOC	0.07 (0.04)	Soil N	0.06 (0.35)	0.59
Cutin monomers	Ambient	$p < 0.05$	Intercept	1.01*** (0.06)	Fine-root biomass	-0.001* (0.000)	--	--	--	--	0.25
	Elevated	$p > 0.05$	Intercept	-5.27* (1.84)	Water table depth	-0.30* (0.10)	Fine-root biomass	0.21* (0.07)	ANPP	0.71* (0.24)	0.44
Suberin monomers	Ambient	$p < 0.05$	Intercept	1.81** (0.42)	Water table depth	0.06* (0.02)	Fine-root biomass	-0.002 (0.002)	--	--	0.45
	Elevated	$p < 0.01$	Intercept	-8.63** (2.92)	Soil N	0.15* (0.06)	SOC	0.02** (0.01)	Fine-root biomass	0.002** (0.001)	0.72
Pyrogenic carbon	Ambient	$p > 0.05$	Intercept	-0.77 (3.91)	Soil N	0.64 (0.33)	--	--	--	--	0.16
	Elevated	$p < 0.05$	Intercept	-0.96 (2.07)	Soil N	0.70** (0.20)	--	--	--	--	0.45
Lignin phenols	Ambient	$p < 0.01$	Intercept	71.11** (16.77)	SOC	-0.14** (0.04)	NPP	0.05** (0.01)	Water table depth	0.38*** (0.06)	0.80
	Elevated	$p < 0.01$	Intercept	56.87**** (7.42)	SOC	-0.06** (0.02)	Water table depth	-0.12** (0.03)	ANPP	0.01 (0.01)	0.72

Comment 14: Line 185 – what do you mean by disproportionately lost? Lignin concentration increased with warming. I am also still left to wonder, which (non-measured) compounds increased with warming, if SOC overall did not change as stated in the Methods?

Response: As previously mentioned in **Comment 6** (Lines 99-104), slowly cycling compounds (e.g., lignin) can increase in relative abundance if more labile compounds decrease in concentration (dilution effect). Since lignin decreased with increasing SOC (Figure 4), we posit that the apparent increase in lignin phenols may be due to dilution by other SOC components such as microbial-processed degradation products. We have modified this sentence it now reads “However, the expectation that the enzymatic breakdown of complex SOC molecules is more sensitive to temperature in comparison to simple molecules³⁴, based in part on the kinetic theory, is not consistent with our results. We show rapid turnover of SOC molecules irrespective of their origins and potential decomposition rates (Fig. 1; Fig. S2).” (Line 206-209).

Comment 15: Line 210 – to show that they are occurring at the same rate, wouldn't you need to discuss the slopes? This might be something helpful to add to the figures that are cited.

Response: For this and for reviewer 2 comment on rates of change, we have added a Supplementary Table 3 with slope estimate for each of the SOC components.

Comment 16: Line 272 – there was no effect of the treatments on SOC concentration? This is surprising given the statement in lines 85-86 that all compounds are vulnerable to loss under warming. If all compounds decline, how is total SOC unchanged?

Response: The difficulties involved in detecting changes in the total size of soil C stocks (Lu et al., 2013; Bradford et al., 2016; Melillo et al., 2017; Song et al., 2019) encouraged the use of molecular compounds comprising SOC, to understand warming effects. However, the issues with using such techniques to infer change in stock sizes echoes those for decomposition; environmental change can alter the sizes of individual C pools or fluxes without altering the total stock (Lu et al., 2013; Bradford et al., 2016; Melillo et al., 2017). Throughout the manuscript now, based on reviewer comments, we have made it clearer where we are referring to SOC molecular components (that are best suited to infer mechanisms and not SOC stocks) vs where we are referring to SOC concentration data derived from solvent-extractable compounds, hydrolysable biopolymers, lignin phenols, and pyrogenic carbon (see methods section) vs where we are referring to other studies like Hanson et al. (2020) that have done an in depth (and independent to our study) analysis of SOC stocks at the same study site.

This sentence now reads “Overall, SOC concentration increased with increasing depth from 44.0% at the surface to 52.5% at 2 m depth, but there was no effect of temperature or eCO₂ on the SOC concentration⁴³. However, investigations of SOC stock at the site so far have shown that the mean effect of warming on soil C stocks was indistinguishable from zero¹⁷. Our observations are not contradictory with this result since the specific compounds we followed represent 20% of the total SOC (see Fig. S1) and should be seen as tracers of SOC.” (Line 296-301).

We have also modified our statement in lines 85-86 that all compounds are vulnerable to loss under warming. This sentence now reads “Overall, our study found support for this hypothesis wherein all molecular compounds comprising SOC, regardless of source and complexity, were vulnerable to shifts in climate drivers, demonstrating the high sensitivity of peatlands to climate change.” (Line 90-92).

Comment 17: Line 318 – there are some known limitations of the CuO method for measuring lignin in soil (for instance, the two references below). Since in this study, you are assessing treatment effects on otherwise identical soils, I don't see a major issue as relative differences should be robust, but do think you should acknowledge that the method often underestimates lignin overall.

Hernes, P. J., Kaiser, K., Dyda, R. Y. & Cerli, C. Molecular trickery in soil organic matter: hidden lignin. *Environ. Sci. Technol.* 47, 9077–9085 (2013).

Klotzbücher, T., Kalbitz, K., Cerli, C., Hernes, P. J. & Kaiser, K. Gone or just out of sight? The apparent disappearance of aromatic litter components in soils. *SOIL* 2, 325–335 (2016).

Response: We agree that an impediment to lignin phenol analysis is that copper (II) oxide (CuO) oxidation method may 'hide' lignin and significantly reduce the yield of lignin phenols upon CuO oxidation (Hernes et al., 2013; Klotzbücher et al., 2016). As demonstrated by Hernes et al. (2013), sorption to iron oxides and minerals may significantly reduce the yield of lignin phenols upon CuO oxidation. We add the following text “We note that while the CuO method is not interfered by any other organic component in peat soils, the method may significantly reduce the yield of phenolic units upon CuO oxidation^{54,55}.” (Line 370-372).

Comment 18: Table S1 – why do the Elevated CO₂ plots tend to have higher soil temperatures within the same nominal warming class?

Response: A key reason for plot-to-plot variation in achieved temperatures and CO₂ concentrations is variable cover by the existing or treatment modified (in some cases) shrub and tree canopies. This variability leads to different energy balance conditions within each enclosure and impacts how diurnal and season temperatures are experienced by the vegetation and microbial communities at depths (Hanson et al., 2017, 2020).

Comment 19: Figure S7 – it's nice to see the soil bulk density data since compound abundances are presented as stocks. Please briefly address how and when bulk density was measured in the Methods.

Response: Thank you for pointing this out. We have added the following statement to address how and when bulk density was measured. “Bulk density measurements were taken from soil cores taken from each of the experimental plots in 2013 and 2020. Soil cores 5.2 cm in diameter were carefully excavated using a Russian corer and soil bulk density was calculated using the freeze-dried weights of the volumetric slices. Bulk density was used to estimate the mass of SOC molecular components (stocks) in the top 30 cm (g/m²). We calculated the mass of SOC molecular components by multiplying bulk density value (g soil cm⁻³) by concentrations of individual SOC compounds (solvent-extractable compounds, hydrolysable biopolymers, lignin phenols and pyrogenic C) (mg g peat⁻¹) from each peat depth.” (Line 301-307).

Comment 20: Data access – when submitting a revised version, will you have submitted datasets and thus can provide a DOI? This would be ideal. If not, can you describe how interested parties can find your datasets within the named repositories?

Response: Yes. Data will be openly available through the SPRUCE repository. We will provide the DOI as soon as the paper is accepted.

Reviewer #2 (Remarks to the Author):

This manuscript presents results on the effect of warming and elevated atmospheric CO₂ concentrations in experimental enclosures in a peatland in Minnesota, over a four-year period. The study provides strong evidence of changes in the chemical composition of the upper layers of the peat associated with both warming and elevated CO₂; the former leads to decreases in selected organic components, associated with enhanced decomposition, while the doubling of CO₂ concentration increased components associated with plant growth, such as roots and leaves/needles. The manuscript is generally well written and structured and presents evidence of possible changes in organic soil components resulting from ‘climate change’. The SPRUCE study was designed to determine the effects of warming and elevated CO₂ concentration, with a clear recognition that the experimental manipulations (warming up to 9°C and CO₂ concentration increased by about 300 ppm) do not represent ‘real’ climate change but give some indication of the expected changes in direction, rather than in magnitude. SPRUCE has been very successful in its operation and the wide range of expertise that have been assembled at the site.

However, I do think that the results, although impressive, need to be presented with some caution: the environmental changes occurred more or less overnight, the range of change is beyond what might be expected to occur (though 2023 may seem to be a portent of rapid changes...), the fall in water table may be moderated by lateral flow in open systems and bogs have many adaptive mechanisms which has allowed them to survive changes in the past. Thus, I suggest the authors tone down some of the statements made, less dramatic and more cautionary in whether the changes shown at SPRUCE will occur in the vast area of northern bogs. The cautionary tone appears in lines 216-219 and it might be applied in the text. Particularly important is the recognition that increases in some SOC components (suberin) are linked to below-ground plant activities (roots), which are expected to have a slower turnover time, with vegetation responding positively to a warmer and drier peat soil.

Response: We thank Prof. Tim Moore for his constructive review as well as for his positive inputs on how to additionally improve our manuscript and we believe these revisions have strongly improved the coherence of the results. We have streamlined the results and discussion sections of the manuscript, by focusing more on the strengths of these responses and discussing the meanings indicated by our measurements. We have also reduced focus on broad extrapolations and toned down the strength of language throughout.

In the future, we hope to complement our results with long-term, time-resolved analysis to see further changes that will have occurred and whether the current findings are generalizable.

I provide some specific comments, by line #:

Comment 1: Line 1 – ‘rapidly’ seems a bit strong given the heavy jolt the system was exposed to and the short duration of the study (4 years temperature change, 2 years CO₂ change).

Response: Also following reviewer 1’s comment, we have removed our use of “rapidly” from the manuscript, including from the title. The title now reads “Climate warming and elevated CO₂ alter peatland soil carbon sources and stability.” (Line 1).

Comment 2: Line 26 – I prefer ‘may’ rather than ‘will’, and it seems that the lowering of the water table may be a significant contributor to the observe changes.

Response: Revised as suggested. This sentence now reads “Together, our results indicate that climate change drivers may increase inputs and enhance the decomposition of SOC potentially destabilising C storage in peatlands” (Line 29-30).

Comment 3: Line 65 – What would be the timescale of turnover of these SOC components in northern peat soils: the components are placed in three cycling groupings.

Response: Turnover of organic matter in peat sequences has been rarely described. It is rather assumed that a majority of the newly formed organic matter in peat is preserved through “humification” processes (Tfaily et al., 2014; Zaccone et al., 2018). Some studies refer to the selective preservation of biomacromolecules (González et al., 2003). Due to the anoxic conditions and the low temperature in the northern peat soil, turnover of these SOC components can be assumed to be much slower than that of well aerated soils that is mostly studied (Schmidt et al., 2011), probably on the centennial timescale in the acrotelm (Hobbie et al., 2016).

Comment 4: Line 72 – The planned increase of 500 ppm CO₂ seems not to have been achieved, from Table S1, more like 300 pp increase.

Response: The experiment only seeks to achieve +500 ppm CO₂ during active seasons and daytime periods (Hanson et al., 2017). In Table S1 the calculated mean CO₂ levels are low because they were averaged from day- and night-time CO₂ data. Also, a key reason for plot-to-plot variation in achieved temperatures and CO₂ concentrations is variable cover by the existing or treatment modified (in some cases) shrub and tree canopies. This variability leads to different energy balance conditions within each enclosure and impacts how diurnal and season temperatures are experienced by the vegetation and microbial communities at depths (Hanson et al., 2017, 2020).

Comment 5: Line 93 – The title implies speed of change and one way to indicate that is to express the rate of change in the SOC components not just as a function of the change in temperature but over the four years of warming. Fig. 1a shows a regression in which there appears to be a decline in ‘solvent-extractable compounds’ from about 23 mg/g at ambient temperature (4oC), to about 16 mg/g at 11.4oC (9oC warming treatment): a decline of about 7 mg/g or about 30% of original content over 4 years. Thus, at the warmest treatment, that would amount to an average loss of about 8% per year (over the four years) and at a more ‘realistic’ warming treatment of 2.25oC (resulting in a soil temperature of about 5.5oC by eyeballing Fig. 1a) of about 1.5 mg/g or 6% of the original over four years, or 1.5% per year. Of course, the changes may be fast at the outset and slow down.

I am not sure whether the above is correct (happy to be corrected), but I wonder how the figure of 7.6 mg/g decrease in solvent-extractable compounds per oC increase in temperature was derived. If the temperature rose by 7oC, then all the original components would be gone. Please correct me if I have this wrong.

Response: Corrected- thank you for pointing out this mistake! We have corrected the change in ‘solvent-extractable compounds’ to $-0.79 \pm 0.2 \text{ mg g}^{-1}$ per degree Celsius increase in temperature over four years. In the regression we compare the temperature responses (rate of change) between +0, +2.25, +4.5, +6.75, and +9 °C plots assuming the plots were at equilibrium at the beginning of the experiment (Tfaily et al., 2014). However, this is a dynamic system in that our observed decrease in SOC components may reflect a transient adjustment period to the applications of the experimental treatments, where our system has not yet reached a new equilibrium.

We liked your suggestion of reporting percent changes between the 0 and 9 °C treatments and now provide two metrics of temperature responses 1) percent change between 0 and 9 (representing the max change; albeit for an extreme temperature scenario and 2) a per degree Celsius change as was previously reported (for example in line 98-100, 101-103, 105-106, 135-136). We have also added a Supplementary Table 3 with slope estimate for each of the SOC components. Furthermore, we have edited rapidly out of the title and out of the manuscript and focus more on the strengths of these responses.

Comment 6: Line 112 – It seems that the lowering of the water table could be a significant response to the warming and elevated CO₂, in terms of the increase in the aerobic zone, possible increase in available nutrients through accelerated decomposition and potential increase in rooting zone. The only metric cited appears to be the maximum depth of the water table over three years, which occurs in August-September. The maximum depth increases from 17 cm to 44 cm (Table S1), a lowering of 27 cm. Is there a more useful metric to indicate the drying of the peat, from the available data? There are few warming experiments of the scale of SPRUCE, but quite a few studies have examined the effect of drainage on peat chemistry and vegetation patterns.

Response: We agree that lowering of the water table does change the subsurface environment from anaerobic to aerobic (oxygenated) at the water table boundary. Such changes will alter the activity of the microbial community and change the mesotelm zone into aerobic functions. The sharp boundary for the water table was documented early in the project and was found to be a clear boundary between aerobic and anaerobic conditions (Tfaily et al., 2014, 2018). As such, we did not invest in depth specific oxygen measurements. Surface moisture measurements are collected at hollow bottoms and for the 0-20 cm peat depth but are quantitatively problematic due to the strong influence of peat bulk density variation near the surface. Nevertheless, they verify unsaturated peat conditions above the water table with strong precipitation events representing the only transient perturbation to this pattern.

Comment 7: Line 160 – Reference might be made to Table S1 in which fine root biomass is presented and shows a six-fold increase from the ‘control’ to the +9 plots (55 and 327 g/m², respectively) and there seems to be a general increase among the plots.

Response: Thank you for pointing this out. To date, findings at SPRUCE experiment suggest that warming and associated peat drying have increased above-ground growth in shrubs and fine-root growth (McPartland et al., 2019; Hanson et al., 2020; Malhotra et al., 2020). We have added the following statement “Importantly, root-derived C (inferred from suberin monomers) was positively correlated with SOC, fine root biomass and water table depth (Fig. 4) and fine root biomass significantly increased under warming and eCO₂ (Table S1)¹⁶, implying increased root-derived C inputs within the surface peat. In corroboration of our results, litter manipulation experiments showed root-derived organic matter to be a source of SOC with greater relative stability and longer turn-over times²⁸, whereas leaf-derived C was found to be turned over more rapidly in (mineral) soils^{27,36,46}.” (Line 186-191).

Comment 8: The data are presented primarily as concentrations per gram of peat, though in Fig. S2 an estimate is made of the mass of components in the top 30 cm. Is there any evidence of the loss of SOC (e.g. g/m²) from the four years of treatments or over a longer period (e.g. the 2012 and 2020 bulk density data in the Supplementary Information)?

Response: Previous studies at the site report no evidence for reductions in soil C stocks based on Hanson et al. (2020). Also see responses above to reviewer 1 **general comment** and **comment 9** and **16** on SOC stocks. We have also added new methods section to address how and when bulk density was measured and estimate of the mass of components in the top 30 cm. “Bulk density measurements were taken from

soil cores taken from each of the experimental plots in 2013 and 2020. Soil cores 5.2 cm in diameter were carefully excavated using a Russian corer and soil bulk density was calculated using the freeze-dried weights of the volumetric slices. Bulk density was used to estimate the mass of SOC molecular components (stocks) in the top 30 cm (g/m^2). We calculated the mass of SOC molecular components by multiplying bulk density value (g soil cm^{-3}) by concentrations of individual SOC compounds (solvent-extractable compounds, hydrolysable biopolymers, lignin phenols and pyrogenic C) (mg g peat^{-1}) from each peat depth.” (Line 301-307).

Comment 9: The components measured (Fig. 1&3) contribute a total of about 95 mg per gram of peat in the ambient plots. How might the remaining 90% of the peat react to the changes?

Response: The main source of SOC is plant- and microbe-derived organic matter (Miltner et al., 2012; Cotrufo et al., 2015; Kallenbach et al., 2016). Here, we investigate SOC molecules defined by organic matter origins (plant-, microorganism- and fire-derived). These molecules carry information on sources and transformation process and are typically regarded to be representative of SOC components of similar origins (Kögel-Knabner, 2002; Otto et al., 2005; Angst et al., 2016; Jansen and Wiesenberger, 2017). We expect therefore the remaining peat to react similarly to the changes reported here.

Comment 10: Fig. 1. I assume the regression is driven through the data for all three depths, and there is little differentiation among the 0-10 cm, 10-20 cm and 20-30 cm depths.

Response: General linear mixed-effect models were used to determine the effects of temperature, elevated CO_2 concentrations and peat depth on individual SOC compounds (solvent-extractable compounds, hydrolysable biopolymers, lignin phenols and pyrogenic C). The depth effect was weakly significant for solvent-extractable compounds (10-20 and 20-30 cm depth) and suberin monomers (20-30 cm depth) under ambient CO_2 treatment and for pyrogenic C (20-30 cm depth) and hydrolysable biopolymers (20-30 cm depth) under elevated CO_2 treatment.

Tim Moore

References

- Angst, G., John, S., Müller, C.W., Kögel-Knabner, I., Rethemeyer, J., 2016. Tracing the sources and spatial distribution of organic carbon in subsoils using a multi-biomarker approach. *Scientific Reports* 6. doi:10.1038/srep29478
- Bailey, V.L., Bond-Lamberty, B., DeAngelis, K., Grandy, A.S., Hawkes, C. V, Heckman, K., Lajtha, K., Phillips, R.P., Sulman, B.N., Todd-Brown, K.E.O., Wallenstein, M.D., 2018. Soil carbon cycling proxies: Understanding their critical role in predicting climate change feedbacks. *Global Change Biology* 24, 895–905. doi:10.1111/gcb.13926
- Bradford, M.A., Wieder, W.R., Bonan, G.B., Fierer, N., Raymond, P.A., Crowther, T.W., 2016. Managing uncertainty in soil carbon feedbacks to climate change. *Nature Climate Change* 6, 751.
- Bragazza, L., Parisod, J., Buttler, A., Bardgett, R.D., 2013. Biogeochemical plant-soil microbe feedback in response to climate warming in peatlands. *Nature Climate Change* 3, 273–277. doi:10.1038/nclimate1781
- Chaudhary, N., Westermann, S., Lamba, S., Shurpali, N., Sannel, A.B.K., Schurgers, G., Miller, P.A., Smith, B., 2020. Modelling past and future peatland carbon dynamics across the pan-Arctic. *Global Change Biology* 26, 4119–4133. doi:10.1111/GCB.15099
- Cheng, L., Zhang, N., Yuan, M., Xiao, J., Qin, Y., Deng, Y., Tu, Q., Xue, K., Van Nostrand, J.D., Wu, L., He, Z., Zhou, X., Leigh, M.B., Konstantinidis, K.T., Schuur, E.A.G., Luo, Y., Tiedje, J.M., Zhou, J., 2017. Warming enhances old organic carbon decomposition through altering functional microbial communities. *ISME Journal* 11, 1825–1835. doi:10.1038/ismej.2017.48
- Cotrufo, M.F., Soong, J.L., Horton, A.J., Campbell, E.E., Haddix, M.L., Wall, D.H., Parton, W.J., 2015. Formation of soil organic matter via biochemical and physical pathways of litter mass loss. *Nature Geoscience* 2015 8:10 8, 776–779. doi:10.1038/ngeo2520
- Feng, X.J., Simpson, A.J., Wilson, K.P., Williams, D.D., Simpson, M.J., 2008. Increased cuticular carbon sequestration and lignin oxidation in response to soil warming. *Nature Geoscience* 1, 836–839. doi:10.1038/ngeo361
- Fenner, N., Freeman, C., 2011. Drought-induced carbon loss in peatlands. *Nature Geoscience* 2011 4:12 4, 895–900. doi:10.1038/ngeo1323
- Freeman, C., Ostle, N., Kang, H., 2001. An enzymic “latch” on a global carbon store: A shortage of oxygen locks up carbon in peatlands by restraining a single enzymes. *Nature* 409, 149. doi:10.1038/35051650
- González, J.A., González-Vila, F.J., Almendros, G., Zancada, M.C., Polvillo, O., Martín, F., 2003. Preferential accumulation of selectively preserved biomacromolecules in the humus fractions from a peat deposit as seen by analytical pyrolysis and spectroscopic techniques. *Journal of Analytical and Applied Pyrolysis* 68–69, 287–298. doi:10.1016/S0165-2370(03)00069-X
- Griffiths, N.A., Hanson, P.J., Ricciuto, D.M., Iversen, C.M., Jensen, A.M., Malhotra, A., McFarlane, K.J., Norby, R.J., Sargsyan, K., Sebestyen, S.D., Shi, X., Walker, A.P., Ward, E.J., Warren, J.M., Weston, D.J., 2017. Temporal and spatial variation in peatland carbon cycling and implications for interpreting responses of an ecosystem-scale warming experiment. *Soil Science Society of America Journal* 0, 0. doi:10.2136/sssaj2016.12.0422
- Hanson, P.J., Griffiths, N.A., Iversen, C.M., Norby, R.J., Sebestyen, S.D., Phillips, J.R., Chanton, J.P., Kolka, R.K., Malhotra, A., Oleheiser, K.C., Warren, J.M., Shi, X., Yang, X., Mao, J., Ricciuto, D.M., 2020. Rapid net carbon loss from a whole-ecosystem warmed peatland. *AGU Advances* 1. doi:10.1029/2020AV000163

- Hanson, P.J., Riggs, J.S., Robert Nettles, W., Phillips, J.R., Krassovski, M.B., Hook, L.A., Gu, L., Richardson, A.D., Aubrecht, D.M., Ricciuto, D.M., Warren, J.M., Barbier, C., 2017. Attaining whole-ecosystem warming using air and deep-soil heating methods with an elevated CO₂ atmosphere. *Biogeosciences* 14, 861–883. doi:10.5194/bg-14-861-2017
- Harwood, J.L., Russell, N.J., 1984. *Lipids in plants and microbes* 162.
- Hernes, P.J., Kaiser, K., Dyda, R.Y., Cerli, C., 2013. Molecular trickery in soil organic matter: Hidden lignin. *Environmental Science and Technology* 47, 9077–9085. doi:10.1021/ES401019N
- Hobbie, E.A., Chen, J., Hanson, P.J., Iversen, C.M., McFarlane, K.J., Thorp, N.R., Hofmockel, K.S., 2016. Long-term carbon and nitrogen dynamics at SPRUCE revealed through stable isotopes in peat profiles. *Biogeosciences Discussions* 1–23. doi:10.5194/bg-2016-261
- Ise, T., Dunn, A.L., Wofsy, S.C., Moorcroft, P.R., 2008. High sensitivity of peat decomposition to climate change through water-table feedback. *Nature Geoscience* 2008 1:11 1, 763–766. doi:10.1038/ngeo331
- Jansen, B., Wiesenberg, G.L.B., 2017. Opportunities and limitations related to the application of plant-derived lipid molecular proxies in soil science. *Soil* 3, 211–234. doi:10.5194/soil-3-211-2017
- Jia, J., Cao, Z., Liu, C., Zhang, Z., Lin, L., Wang, Y., Haghypour, N., Wacker, L., Bao, H., Dittmar, T., Simpson, M.J., Yang, H., Crowther, T.W., Eglinton, T.I., He, J.-S., Feng, X., 2019. Climate warming alters subsoil but not topsoil carbon dynamics in alpine grassland. *Global Change Biology* 25, 4383–4393. doi:10.1111/GCB.14823
- Kallenbach, C.M., Frey, S.D., Grandy, A.S., 2016. Direct evidence for microbial-derived soil organic matter formation and its ecophysiological controls. *Nature Communications* 2016 7:1 7, 1–10. doi:10.1038/ncomms13630
- Klotzbücher, T., Kalbitz, K., Cerli, C., Hernes, P.J., Kaiser, K., 2016. Gone or just out of sight? The apparent disappearance of aromatic litter components in soils. *Soil* 2, 325–335. doi:10.5194/soil-2-325-2016
- Kögel-Knabner, I., 2002. The macromolecular organic composition of plant and microbial residues as inputs to soil organic matter. *Soil Biology and Biochemistry*. doi:10.1016/S0038-0717(01)00158-4
- Kramer, C., Gleixner, G., 2006. Variable use of plant- and soil-derived carbon by microorganisms in agricultural soils. *Soil Biology and Biochemistry* 38, 3267–3278. doi:10.1016/j.soilbio.2006.04.006
- Liu, L., Chen, H., Tian, J., 2022. Varied response of carbon dioxide emissions to warming in oxic, anoxic and transitional soil layers in a drained peatland. *Communications Earth & Environment* 2022 3:1 3, 1–12. doi:10.1038/s43247-022-00651-y
- Lu, M., Zhou, X., Yang, Q., Li, H., Luo, Y., Fang, C., Chen, J., Yang, X., Li, B., 2013. Responses of ecosystem carbon cycle to experimental warming: a meta-analysis. *Ecology* 94, 726–738. doi:10.1890/12-0279.1
- Malhotra, A., Brice, D.J., Childs, J., Graham, J.D., Hobbie, E.A., Vander Stel, H., Feron, S.C., Hanson, P.J., Iversen, C.M., 2020. Peatland warming strongly increases fine-root growth. *Proceedings of the National Academy of Sciences* 117, 202003361. doi:10.1073/pnas.2003361117
- McPartland, M.Y., Montgomery, R.A., Hanson, P.J., Phillips, J.R., Kolka, R., Palik, B., 2019. Vascular plant species response to warming and elevated carbon dioxide in a boreal peatland. *Environmental Research Letters* 15, 124066. doi:10.1088/1748-9326/abc4fb
- Melillo, J.M., Frey, S.D., DeAngelis, K.M., Werner, W.J., Bernard, M.J., Bowles, F.P., Pold, G., Knorr, M.A., Grandy, A.S., 2017. Long-term pattern and magnitude of soil carbon feedback to the climate system in a warming world. *Science* 358, 101–105. doi:10.1126/science.aan2874

- Miltner, A., Bombach, P., Schmidt-Brücken, B., Kästner, M., 2012. SOM genesis: Microbial biomass as a significant source. *Biogeochemistry* 111, 41–55. doi:10.1007/S10533-011-9658-Z/METRICS
- Ofiti, N.O.E., Solly, E.F., Hanson, P.J., Malhotra, A., Wiesenberg, G.L.B., Schmidt, M.W.I., 2022. Warming and elevated CO₂ promote rapid incorporation and degradation of plant-derived organic matter in an ombrotrophic peatland. *Global Change Biology* 28, 883–898. doi:10.1111/GCB.15955
- Otto, A., Shunthirasingham, C., Simpson, M.J., 2005. A comparison of plant and microbial biomarkers in grassland soils from the prairie ecozone of Canada. *Organic Geochemistry* 36, 425–448. doi:10.1016/j.orggeochem.2004.09.008
- Pisani, O., Frey, S.D., Simpson, A.J., Simpson, M.J., 2015. Soil warming and nitrogen deposition alter soil organic matter composition at the molecular-level. *Biogeochemistry* 123, 391–409. doi:10.1007/s10533-015-0073-8
- Pold, G., Billings, A.F., Blanchard, J.L., Burkhardt, D.B., Frey, S.D., Melillo, J.M., Schnabel, J., van Diepen, L.T.A., DeAngelis, K.M., 2016. Long-term warming alters carbohydrate degradation potential in temperate forest soils. *Applied and Environmental Microbiology* 82, 6518–6530. doi:10.1128/AEM.02012-16
- Schmidt, M.W.I., Torn, M.S., Abiven, S., Dittmar, T., Guggenberger, G., Janssens, I.A., Kleber, M., Kögel-Knabner, I., Lehmann, J., Manning, D.A.C., Nannipieri, P., Rasse, D.P., Weiner, S., Trumbore, S.E., 2011. Persistence of soil organic matter as an ecosystem property. *Nature* 478, 49.
- Song, J., Wan, S., Piao, S., Knapp, A.K., Classen, A.T., Vicca, S., Ciais, P., Hovenden, M.J., Leuzinger, S., Beier, C., Kardol, P., Xia, J., Liu, Q., Ru, J., Zhou, Z., Luo, Y., Guo, D., Adam Langley, J., Zscheischler, J., Dukes, J.S., Tang, J., Chen, J., Hofmockel, K.S., Kueppers, L.M., Rustad, L., Liu, L., Smith, M.D., Templer, P.H., Quinn Thomas, R., Norby, R.J., Phillips, R.P., Niu, S., Faticchi, S., Wang, Y., Shao, P., Han, H., Wang, D., Lei, L., Wang, J., Li, Xiaona, Zhang, Q., Li, Xiaoming, Su, F., Liu, B., Yang, F., Ma, G., Li, G., Liu, Yanchun, Liu, Yinzhan, Yang, Z., Zhang, K., Miao, Y., Hu, M., Yan, C., Zhang, A., Zhong, M., Hui, Y., Li, Y., Zheng, M., 2019. A meta-analysis of 1,119 manipulative experiments on terrestrial carbon-cycling responses to global change. *Nature Ecology and Evolution* 3, 1309–1320. doi:10.1038/s41559-019-0958-3
- Tfaily, M.M., Cooper, W.T., Kostka, J.E., Chanton, P.R., Schadt, C.W., Hanson, P.J., Iversen, C.M., Chanton, J.P., M., T.M., T., C.W., E., K.J., R., C.P., W., S.C., J., H.P., M., I.C., P., C.J., Tfaily, M.M., Cooper, W.T., Kostka, J.E., Chanton, P.R., Schadt, C.W., Hanson, P.J., Iversen, C.M., Chanton, J.P., 2014. Organic matter transformation in the peat column at Marcell experimental forest: Humification and vertical stratification. *Journal of Geophysical Research: Biogeosciences* 119, 661–675. doi:10.1002/2013JG002492
- Tfaily, M.M., Wilson, R.M., Cooper, W.T., Kostka, J.E., Hanson, P., Chanton, J.P., 2018. Vertical stratification of peat pore water dissolved organic matter composition in a peat bog in northern Minnesota. *Journal of Geophysical Research: Biogeosciences* 123, 479–494. doi:10.1002/2017JG004007
- Walker, T.N., Garnett, M.H., Ward, S.E., Oakley, S., Bardgett, R.D., Ostle, N.J., 2016. Vascular plants promote ancient peatland carbon loss with climate warming. *Global Change Biology* 22, 1880–1889.
- Wang, H., Richardson, C.J., Ho, M., 2015. Dual controls on carbon loss during drought in peatlands. *Nature Climate Change* 5:6 5, 584–587. doi:10.1038/nclimate2643
- Wiesenberg, G.L.B., Schwarzbauer, J., Schmidt, M.W.I., Schwark, L., 2008. Plant and soil lipid modification under elevated atmospheric CO₂ conditions: II. Stable carbon isotopic values ($\delta^{13}\text{C}$) and turnover. *Organic Geochemistry* 39, 103–117. doi:10.1016/j.orggeochem.2007.09.006

- Wilson, R.M., Griffiths, N.A., Visser, A., McFarlane, K.J., Sebestyen, S.D., Oleheiser, K.C., Bosman, S., Hopple, A.M., Tfaily, M.M., Kolka, R.K., Hanson, P.J., Kostka, J.E., Bridgham, S.D., Keller, J.K., Chanton, J.P., 2021a. Radiocarbon analyses quantify peat carbon losses with increasing temperature in a whole ecosystem warming experiment. *Journal of Geophysical Research: Biogeosciences* 126, e2021JG006511. doi:10.1029/2021JG006511
- Wilson, R.M., Hough, M.A., Verbeke, B.A., Hodgkins, S.B., Tyson, G., Sullivan, M.B., Brodie, E., Riley, W.J., Woodcroft, B., McCalley, C., Dominguez, S.C., Crill, P.M., Varner, R.K., Frolking, S., Cooper, W.T., Chanton, J.P., Saleska, S.D., Rich, V.I., Tfaily, M.M., 2022. Plant organic matter inputs exert a strong control on soil organic matter decomposition in a thawing permafrost peatland. *Science of The Total Environment* 820, 152757. doi:10.1016/J.SCITOTENV.2021.152757
- Wilson, R.M., Tfaily, M.M., Kolton, M., Johnston, E.R., Petro, C., Zalman, C.A., Hanson, P.J., Heyman, H.M., Kyle, J.E., Hoyt, D.W., Eder, E.K., Purvine, S.O., Kolka, R.K., Sebestyen, S.D., Griffiths, N.A., Schadt, C.W., Keller, J.K., Bridgham, S.D., Chanton, J.P., Kostka, J.E., 2021b. Soil metabolome response to whole-ecosystem warming at the spruce and peatland responses under changing environments experiment. *Proceedings of the National Academy of Sciences of the United States of America* 118. doi:10.1073/PNAS.2004192118
- Zaccone, C., Plaza, C., Ciavatta, C., Miano, T.M., Shotyk, W., 2018. Advances in the determination of humification degree in peat since : Applications in geochemical and paleoenvironmental studies. *Earth-Science Reviews* 185, 163–178. doi:10.1016/J.EARSCIREV.2018.05.017
- Zosso, C.U., Ofiti, N.O.E., Torn, M.S., Wiesenberg, G.L.B., Schmidt, M.W.I., 2023. Rapid loss of complex polymers and pyrogenic carbon in subsoils under whole-soil warming. *Nature Geoscience* 2023 1–5. doi:10.1038/s41561-023-01142-1

REVIEWERS' COMMENTS

Reviewer #1 (Remarks to the Author):

I reviewed a previous version of this manuscript. I appreciate the author's work to revise the text and find it much improved. However, I think there remain areas where the language can be clarified to improve precision of their arguments and ensure readers can follow the methods, key findings, and proposed rationale.

Lines 27-29 – I wonder if the second to last sentence of the abstract can be re-formulated. 'The observed decline in SOC' sounds like stocks, perhaps change to 'SOC molecules'? And the reference to 'younger' makes me expect radiocarbon, is this word necessary?

Lines 53-54 – 'composition and activity of the microbial community' could you mention enzymes since the oxidative enzymes are likely a key factor? Perhaps you could add 'and their enzymes' to encompass this?

Line 81 – in the Response to Reviewers doc, you mention this would be a new paragraph but this isn't how it appears in the submission. I think this would better, consider updating.

Line 90 – But the stated hypothesis is that more chemically complex compounds would decompose faster, how did you 'find support' for this if all the compounds decreased? Suggest you acknowledge this nuance in the text.

Lines 118-120 – but lignin was lost less so compared to other types of compounds, which is a bit surprising no? I think it's worth re-iterating this interesting point by amending this sentence to something like, 'Overall, the above findings indicate that warming stimulated a loss of SOC molecules, irrespective of their origins and potential decomposition rates – although lignin declined less relative to other compound classes.' Also, I would consider starting a new paragraph with this sentence, just my opinion but I find shorter paragraphs easier to read and follow.

Line 145 – warming treatments? I'm surprised, do you mean eCO₂? Since this paragraph is about how eCO₂ plus warming responses are different from warming alone.

Line 165 – should 'Nevertheless' be revised to 'Likely because the deeper soils remained waterlogged' or similar? I like the new text about water table effects, but since the drop was about 30 cm (if I'm understanding correctly), it's not surprising that the deep peat was unaffected, since it was not oxygenated like the surface peat, correct?

Line 186 – by 'plant quality' do you mean plant tissue chemistry? Might say this, or at least give a very brief definition of what quality means in this context

Line 220 – Given Figure S1, it seems '13% of the SOC' is incorrect. Maybe 13% of the molecules

examined in this study, but from looking at the Y axis of Figure S1 it looks like only 1-2% of total SOC. Please revise.

Line 298 – it seems like ‘However’ should be ‘Additionally’ since this sentence agrees with the one before, there was no change with treatment.

Lines 305-306 and Figures S2 and S3 – I want to make sure of the units here. To get to g/m² from the starting units indicated (g/cm³, mg/g, cm), I believe you need to multiply by 10. Is that what you did? If so, please mention it or include an equation.

Table S3 – ‘from a single bivariate linear regression’ – with what variables, each compound regressed against temperature? Please clarify in the table caption. It might also be helpful to reference this table in the methods section, perhaps Line 410?

Line 426 – what is the ‘global model’? Not seeing that in Table S4. Also, there are more variables listed in the Predictor column of that table compared to what is listed in the text.

Reviewer #2 (Remarks to the Author):

I have reviewed the response of the authors to my comments (though the format of the text is difficult to decipher in parts). I found the responses are adequate and the manuscript is more 'reasonable' and clearer in parts. I did not evaluate the responses to Reviewer 1.

Reviewers' comments:

Reviewer #1 (Remarks to the Author):

I reviewed a previous version of this manuscript. I appreciate the author's work to revise the text and find it much improved. However, I think there remain areas where the language can be clarified to improve precision of their arguments and ensure readers can follow the methods, key findings, and proposed rationale.

Response: We thank the reviewer for the constructive and insightful review of the earlier version as well as for the positive inputs on how to additionally improve our manuscript. We have now clarified wording where it was unclear as per reviewer suggestions. We now believe that the novelties of our study are strengthened in the revised manuscript.

Comment 1: Lines 27-29 – I wonder if the second to last sentence of the abstract can be re-formulated. 'The observed decline in SOC' sounds like stocks, perhaps change to 'SOC molecules'? And the reference to 'younger' makes me expect radiocarbon, is this word necessary?

Response: We agree and have removed this statement. Based on reviewer suggestions, we have revised this sentence it now reads "The decline in SOC compounds with warming and gains from new root-derived C under eCO₂, suggest that warming and eCO₂ may shift peatland C budget towards pools with faster turnover." (Line 26-28).

Comment 2: Lines 53-54 – 'composition and activity of the microbial community' could you mention enzymes since the oxidative enzymes are likely a key factor? Perhaps you could add 'and their enzymes' to encompass this?

Response: Good point! We have added the following statement to acknowledge the fact that oxidative enzymes are likely a key factor in the microbially-mediated turnover of belowground soil carbon under lower water-tables and increased peat aeration. This sentence now reads "Such a change in peat aeration and plant litter (including root exudates) inputs can impact the microbially-mediated SOC turnover because soil water content and litter chemistry regulate microbial community composition, function, and enzymatic activity⁸⁻¹¹." (Line 51-53).

Comment 3: Line 81 – in the Response to Reviewers doc, you mention this would be a new paragraph, but this isn't how it appears in the submission. I think this would better, consider updating.

Response: Thank you for pointing this out. This section is now a new paragraph (Line 81).

Comment 4: Line 90 – But the stated hypothesis is that more chemically complex compounds would decompose faster, how did you 'find support' for this if all the compounds decreased? Suggest you acknowledge this nuance in the text.

Response: We initially stated this 'support' with the perspective that even the complex compounds responded. However, as per your comment, it indeed makes more sense to state this as a rejected hypothesis. We have now changed the sentence, specifying that our results reject the hypothesis that complex SOC molecules are more responsive to temperature changes as all molecular compounds comprising SOC, regardless of source and complexity, were vulnerable to shifts in climate drivers. We have revised this sentence it now reads "Overall, our study rejects this hypothesis that complex SOC molecules are more responsive to temperature changes. Instead, we found that all molecular compounds

comprising SOC, regardless of source and complexity, were vulnerable to shifts in climate drivers, demonstrating the high sensitivity of peatlands to climate change” (Line 90).

Comment 5: Lines 118-120 – but lignin was lost less so compared to other types of compounds, which is a bit surprising no? I think it’s worth re-iterating this interesting point by amending this sentence to something like, ‘Overall, the above findings indicate that warming stimulated a loss of SOC molecules, irrespective of their origins and potential decomposition rates – although lignin declined less relative to other compound classes.’ Also, I would consider starting a new paragraph with this sentence, just my opinion but I find shorter paragraphs easier to read and follow.

Response: Good point! This sentence was revised as suggested and now reads “Overall, the above findings indicate that warming stimulated a loss of SOC molecules, irrespective of their origins and potential decomposition rates – although lignin phenols showed a lower response than expected.” (Line 120-122). This section is also a new paragraph.

Comment 6: Line 145 – warming treatments? I’m surprised, do you mean eCO₂? Since this paragraph is about how eCO₂ plus warming responses are different from warming alone.

Response: We have revised the sentence and made it clearer that we are talking about warming plus elevated atmospheric CO₂ concentration (eCO₂) treatment. This sentence now reads: “Previous work at the study site demonstrated increased above- and belowground vascular plant biomass in response to warming and eCO₂ treatments^{16,17,20}.” (Line 147).

Comment 7: Line 165 – should ‘Nevertheless’ be revised to ‘Likely because the deeper soils remained waterlogged’ or similar? I like the new text about water table effects, but since the drop was about 30 cm (if I’m understanding correctly), it’s not surprising that the deep peat was unaffected, since it was not oxygenated like the surface peat, correct?

Response: We have removed this statement and changed the formulation to show that surface peat had stronger responses than deeper peat and that this is likely due to the water table. This sentence now reads “Four years into the treatment, deep anaerobic peat appears to be stable (Fig. S5-6) likely due to the slower decomposition that occurs under anoxic conditions^{2,9} and low-quality C substrates in these layers^{13,35}.” (Line 167-169).

Comment 8: Line 186 – by ‘plant quality’ do you mean plant tissue chemistry? Might say this, or at least give a very brief definition of what quality means in this context.

Response: Here, we were referring to specific vegetation chemistry of overstory trees, ericaceous shrubs, and bryophyte layer. Based on reviewer comment, we have removed this statement as we realized that it was tangential to increased Suberin in SOC and corresponding increases in root inputs (Line 184).

Comment 9: Line 220 – Given Figure S1, it seems ‘13% of the SOC’ is incorrect. Maybe 13% of the molecules examined in this study, but from looking at the Y axis of Figure S1 it looks like only 1-2% of total SOC. Please revise.

Response: Corrected- thank you for pointing out this mistake! We have corrected proportion of total pyrogenic carbon normalized to organic carbon concentration in the surface peat to 1.4%. This sentence now reads “We did not observe significant changes in PyC concentrations under ambient CO₂ or eCO₂

($p > 0.05$; Fig. 1c; Fig. S4a), despite these peat soils containing PyC above global averages (~1.4% of the SOC; Fig. S1)⁴⁸.” (Line 218).

Comment 10: Line 298 – it seems like ‘However’ should be ‘Additionally’ since this sentence agrees with the one before, there was no change with treatment.

Response: Revised as suggested. This sentence now reads “Additionally, investigations of SOC stock at the site so far have shown that the mean effect of warming on soil C stocks was indistinguishable from zero¹⁷.” (Line 296).

Comment 11: Lines 305-306 – and Figures S2 and S3 – I want to make sure of the units here. To get to g/m² from the starting units indicated (g/cm³, mg/g, cm), I believe you need to multiply by 10. Is that what you did? If so, please mention it or include an equation.

Response: Good point! We calculated the mass of SOC molecular components by multiplying bulk density value (g soil cm⁻³) by concentrations of individual SOC compounds (g g peat⁻¹) from each peat depth. We have modified this sentence it now reads: “We calculated the mass of SOC molecular components by multiplying bulk density value (g soil cm⁻³) by concentrations of individual SOC compounds (solvent-extractable compounds, hydrolysable biopolymers, lignin phenols and pyrogenic C) (g g peat⁻¹) from each peat depth and summarized the values for the mentioned depth intervals.” (Line 303-306).

Comment 13: Table S3 – ‘from a single bivariate linear regression’ – with what variables, each compound regressed against temperature? Please clarify in the table caption. It might also be helpful to reference this table in the methods section, perhaps Line 410?

Response: For each SOC compound, we evaluated separate regressions against temperature for ambient and elevated CO₂ treatments. We have modified this sentence it now reads: “Each reported slope is from a single bivariate linear regression of each compound regressed against temperature under ambient or elevated CO₂ treatment in the surface peat (0-30 cm depth) following 4 years of warming and 2 years of elevated atmospheric CO₂ concentrations” (see Table S3).

We also reference Table S3 in the methods section (Line 409).

Comment 14: Line 426 – what is the ‘global model’? Not seeing that in Table S4. Also, there are more variables listed in the Predictor column of that table compared to what is listed in the text.

Response: We changed the formulation and hope we can now avoid any confusion. “In the multiple linear regression model, potential explanatory variables included: SOC concentration, nitrogen concentration, net primary productivity, fine-root biomass, and maximum distance to the water table (Fig. 4; Table S4).” (Line 425-427).

Reviewer #2 (Remarks to the Author):

I have reviewed the response of the authors to my comments (though the format of the text is difficult to decipher in parts). I found the responses are adequate and the manuscript is more 'reasonable' and clearer in parts. I did not evaluate the responses to Reviewer 1.

Response: We thank the reviewer for the constructive review of the earlier version.